



# Climatic, weather and socio-economic conditions corresponding with the mid-17th century eruption cluster

Markus Stoffel[1,2,3], Christophe Corona[1,4], Francis Ludlow[5], Michael Sigl[6], Heli Huhtamaa[7], Emmanuel Garnier[8], Samuli Helama[9], Sébastien Guillet[1], Arlene Crampsie[10], Katrin Kleemann[11,12], Chantal Camenisch[7], Joseph McConnell[13], Chaochao Gao[14]

[1] Change Impacts and Risks in the Anthropocene (C-CIA), Institute for Environmental Sciences, University of Geneva, 1205 Geneva, Switzerland
[2] Department of Earth Sciences, University of Geneva, 1205 Geneva, Switzerland
[3] Department F.-A. Forel for Environmental and Aquatic Sciences, University of Geneva, 1205 Geneva, Switzerland
[4] Geolab, Université Clermont Auvergne, CNRS, 63000, Clermont-Ferrand, France
[5] Trinity Centre for Environmental Humanities, and Department of History, Trinity College Dublin, Ireland
[6] Climate and Environmental Physics and Oeschger Centre for Climate Change Research, University of Bern, 3012 Bern, Switzerland
[7] Institute of History and Oeschger Centre for Climate Change Research, University of Bern, 3012 Bern, Switzerland
[8] UMR 6249 CNRS Chrono-Environnement, University of Bourgogne Franche-Comté, France
[9] Natural Resources Institute Finland, Ounasjoentie 6, 96200 Rovaniemi, Finland
[10] School of Geography, University College Dublin, Belfield, Dublin 4, Ireland
[11] Department of History, University of Freiburg, 79085 Freiburg im Breisgau, Germany
[12] German Maritime Museum - Leibniz Institute for Maritime History, 27568 Bremerhaven, Germany
[13] Division of Hydrologic Sciences, Desert Research Institute, 89512, Reno, USA
[14] School of Environmental and Resource Sciences, Zhejiang University, Hangzhou 310058, China

*Correspondence to*: Markus Stoffel (markus.stoffel@unige.ch)

**Abstract.** The mid-17th century is characterized by a cluster of explosive volcanic eruptions in the 1630s and 1640s, deteriorating climatic conditions culminating in the Maunder Minimum as well as political instability and famine in regions of Western and Northern Europe as well as China and Japan. This contribution investigates the sources of the eruptions of the 1630s and 1640s and their possible impact on contemporary climate using ice-core, tree-ring and historical evidence, but will also look into the socio-political context in which they occurred and the human responses they may have triggered. Three distinct sulfur peaks are found in the Greenland ice core record in 1637, 1641-42 and 1646. In Antarctica, only one unambiguous sulfate spike is recorded, peaking in 1642. The resulting bipolar sulfur peak in 1641-1642 can likely be ascribed to the eruption of Mount Parker (6°N, Philippines) on December 26, 1640, but sulfate emitted from Koma-ga-take (42°N, Japan) volcano on July 31, 1641, has potentially also contributed to the sulphate concentrations observed in Greenland at this time. The smaller peaks in 1637 and 1646 can be potentially attributed to the eruptions of Hekla (63°N, Iceland) and Shiveluch (56°N, Russia), respectively. To date, however, none of the candidate volcanoes for the mid-17th century sulphate peaks have been confirmed with tephra preserved in ice cores. Tree-ring and written sources point to severe and cold conditions in the late 1630s and early 1640s in various parts of Europe, and to poor harvests. Yet the early 17th century was also characterized by widespread warfare across Europe – and in particular the Thirty Years' War (1618–1648), rendering any attribution of socio-





economic crisis to volcanism challenging. In China and Japan, historical sources point to extreme droughts and famines starting in the late 1630s, and thus preceding the eruptions by some years. The case of the eruption cluster in the late 1630s and early

1640s and the climatic and societal conditions recorded in its aftermath thus offer a textbook example of difficulties in (i) unambiguously distinguishing volcanically induced cooling, wetting or drying from natural climate variability, and (ii) attributing political instability, harvest failure and famines solely to volcanic climatic impacts. This example shows that the impacts of past volcanism must always be studied within the contemporary socio-economic contexts, but that it is also time to most past reductive framings and sometimes reactionary oppositional stances in which climate (and environment more broadly)

either is or is not deemed an important contributor to major historical events.

## 1 Introduction

The Little Ice Age (or LIA) is a proposed climate interval during which glaciers expanded in mountain regions across the globe (Lamb and Grove, 1989; Matthews and Briffa, 2005), including the European Alps (Holzhauser et al., 2007; Nussbaumer et

al., 2007), New Zealand (Lorrey et al., 2014), Alaska (Wiles et al., 1999), the Northern Rockies (Luckman, 2000), and the southern Andes (Masiokas et al., 2009). Glacier advance was favored by widespread but spatially and temporally heterogeneous cooling (Hegerl et al., 2011; Luterbacher et al., 2016; Ljungqvist et al., 2019; Neukom et al., 2019) of mean annual temperatures on the order of *c.* 0.6°C relative to the last millennium average across the Northern Hemisphere (NH) (Mann et al., 2009).

In this context, the mid- and late-17th century represents an important phase of the LIA – and hence for paleoclimate research – due to the occurrence of very low solar activity from approximately 1621 to 1718 (Usoskin et al., 2014; Owens et al., 2017; Brehm et al., 2021). This phase, referred to as the Maunder Minimum (Eddy, 1976), coincides with a period of cooler temperatures (Luterbacher et al., 2004; Xoplaki et al., 2005) and strong decadal variability in summer and autumn precipitation as of the mid-17th century (Pauling et al., 2006) over Europe. Physicists have thus considered the cooling as compelling

evidence of a large, direct solar influence on climate (Gray et al., 2010). More recent research argues that this temporal agreement alone does not establish physical causation between the solar Maunder Minimum and the Little Ice Age. Instead, multiple factors probably contributed to the cooling, including the drop in solar activity and changes in land use. The largest influence is, however, likely attributable to volcanic eruptions (Crowley, 2000; Owens et al., 2017; Slawinska and Robock, 2018) and their impact on the North Atlantic Oscillation (NAO; Booth et al., 2012; Zanchettin et al., 2013), the subpolar gyre

(SPG; White et al., 2021) and sea ice growth over the Nordic seas (Lehner et al., 2013; Schleussner and Feulner, 2013).

The first few decades of the 17th century also experienced marked political instability and warfare. In Europe, the Thirty Years War (1618–1648), which origins were connected to religious civil wars over German-speaking areas, turned quickly into a geopolitical war affecting most of continental Europe (Parker, 2013; Schmidt, 2018). By the turn of the 1630s and 1640s, the Holy Roman Empire, the Spanish Empire, Denmark–Norway, the Swedish Kingdom, France, and the Dutch Republic were



involved in at least some warfare. Moreover, the period was characterized by civil wars including the Scottish Revolution (1637–1644), the *Croquant* (1637) and the *Nu-pieds* (1639) revolts in France, the Catalan Revolt (1640–1659), the Portugal rebels (1640–1668), the Irish Rebellion (1641–1642) and English Civil War (1642–1651) (Parker 2013). In China, the collapse of the Ming Dynasty (1368-1644), overthrown by a peasant uprising coinciding with severe drought and famine, is yet another superposition of climatic and political events (Zheng et al., 2014; Gao et al., 2021).

This paper aims to shed light on the climatic and socio-economic events that have shaped 17[th] century societies across Europe, China, and Japan, thereby putting the eruption cluster between 1637 and 1646 into context, and particularly the most substantial eruption(s) around 1640. This paper aims to underline how pre-existing weaknesses in socio-economic systems and unstable political conditions have been key to transforming the climatic effects of volcanic eruptions into natural disasters. To this end, we here (i) summarize the current state of knowledge on the mid-17[th] century eruptions from ice cores, (ii) identify – to the

degree possible – the likely source volcanoes, (iii) reconstruct the climatic conditions that prevailed at the time of these eruptions over NH landmasses and within Europe, and (iv) examine weather and climatic anomalies reported by chroniclers in Europe and Asia as well as the socio-political contexts in which these occurred. We conclude that attribution of impacts from the mid-17[th] century eruptions on contemporary societies remains difficult as these events – and their associated volcanic cooling – occurred in a time of an already worsening climate, with pre-existing widespread and marked political instability as

well as diminishing solar activity at the start of the Maunder minimum, though this should not be automatically seen as precluding a role for volcanically induced climatic cooling in the historical events of the period.

## 2 Mid-17[th] century eruptions in the ice-core records

Ice cores from Greenland and Antarctic ice sheets contain information of Earth's past atmospheric composition, including concentrations of acid and/or sulfate emitted by past eruptions (Crowley and Unterman, 2013; Gao et al., 2008; Sigl et al.,

2014; Toohey and Sigl, 2017). Compared to other geological records (Brown et al., 2014), ice cores provide high dating accuracy and precision and offer virtually complete time series of historical explosive eruptions of the size of Pinatubo (1991) or larger (Sigl et al., 2015). The network of available ice cores is dense for the mid-17[th] century, and here we employ sulfate measurements dated according to the annual-layer counted chronology WD2014 for the WAIS Divide (WDC) core (Antarctica) and the NS1-2011 chronology for the NEEM (2011S1) core (Greenland). We additionally use new high-resolution

non-sea-salt sulfur (nssS) measurements corrected for sulfur contained in sea salt from Summit15 (Greenland; 72.6°N, 38.5°W, 3210 m asl), TUNU13 (Greenland; Sigl et al., 2015) and B40 (Antarctica; Sigl et al., 2015). Measurements were obtained by inductively coupled plasma mass spectrometry (ICPMS) conducted at the Desert Research Institute, Reno (USA), using the methodology described by Sigl et al. (2014). Monthly values shown in Fig. 1 were derived by assuming linear snowfall throughout the year.

The period 1630–1650 shows three distinct volcanic sulfur peaks in Greenland with concentrations peaking in 1637, 1641-42, and 1646 (Fig. 1a). A smaller peak observed in NEEM (2011S1) in 1640 is not recorded in the two other ice cores. During the





same period, only one unambiguous sulfur spike is recorded in Antarctica, observed in 1642 in both cores (Fig. 1b). The duration of elevated sulfate concentrations is 0.5 (1637), 2.3 (1641-42) and 0.8 (1646) years in Greenland, and 2.1 (1641-42) years in Antarctica, respectively (Table 1), most likely reflecting differences in the aerosol lifecycle of each eruption (Schmidt

and Robock, 2015). For reference, the mean duration of volcanic sulphate deposition from the same ice core sites is 2.6 years for the eruptions of Tambora in 1815 (Brönnimann et al., 2019) and Samalas in 1257 (Lavigne et al., 2013; Guillet et al., 2017), and 1.7 years for the eruptions of Krakatau in 1883 (Verbeek, 1884; Winchester, 2003) and Pinatubo in 1991 (Parker et al., 1996).

Time-integrated sulfate mass deposition rates ($f$) are 7 yr (1637); 41 yr (1641-42) and 13 kg km$^{-2}$ yr (1646) over the Greenland

ice core sites, and 15 kg km$^{-2}$ yr (1641-42) over the Antarctica ice core sites (Table 1; Toohey and Sigl, 2017). An asymmetry ratio defined as A= $f_{Greenland}$ / ($f_{Greenland}$ + $f_{Antarctica}$) of 0.73 indicates that the stratospheric sulphate burden after the 1641-42 cluster was more strongly confined to NH. This contrasts with other notable stratospheric eruptions in the tropics that produced more symmetrical hemispheric sulfate burdens (e.g., Tambora, Indonesia, 1815: A=0.46; or Samalas, Indonesia, 1257; A=0.59). Even if we attribute the sulfate peak in 1641-42 observed in both polar regions primarily to a tropical eruption, a NH

eruption may also have contributed to increased sulfate concentrations in Greenland. However, this potential contribution cannot presently be quantified given the necessarily close timing of both hypothesized eruptions and the different lifecycle of atmospheric aerosol from tropical and extra-tropical eruptions (Toohey et al., 2013, 2019).

As aerosol model simulations suggest limited transport of volcanic sulfate aerosols into the polar regions of the opposing hemisphere, the two smaller sulfate peaks observed only in the ice cores from Greenland in 1637 and 1646 can likely be

attributed to eruptions in the mid-to-high NH latitudes.

The magnitude of the stratospheric sulfur injection for the eruptions producing the observed ice-core sulfur signals has been estimated at 1.3 (1637), 18.7 (1640/41) and 2.4 (1646) Tg S (Toohey and Sigl, 2017). The temporal clustering of the Parker and Koma-ga-take eruptions (as best present candidates for the 1641/42 signal) thus injected twice as much sulfur as Krakatau (6°S, Indonesia, 1883; 9.3 Tg S), almost as much as Huaynaputina (17°S, Peru, 1600; 18.9 Tg S), and roughly two-thirds of

Tambora (1815; 28.1 Tg S) (Table 1). Forward modelling of atmospheric properties suggests strong increases in atmospheric opacity (i.e., stratospheric optical depth, SAOD) globally and particularly between 30 and 90°N latitude after the eruption cluster (Fig. 1c). Peak SAOD$_{30-90°N}$ values occurred in June 1637 (0.05), Dec 1641 (0.22), and June 1646 (0.08), respectively. By comparison, the Pinatubo eruption had an SAOD$_{30-90°N}$ of 0.12, while Tambora triggered an increase to up to 0.32. Following Hansen et al. (2005), we estimate that peak global radiative forcing (RF) due to the increased SAOD from these

eruptions was –0.5 W m$^{-2}$ (1637), –6.0 W m$^{-2}$ (1641/42) and –1.0 W m$^{-2}$ (1646).

The radiative forcing of the 1640/41 eruption(s) thus ranks among the largest of the past 1000 years, being about twice as large as Pinatubo (1991) or Krakatau (1883), and comparable to Huaynaputina in 1600 (Fig. 2). The eruption in 1646, though substantially smaller than the 1640/41 eruption(s), would still have been a significant eruption had it occurred in the 20[th] century. Its SAOD$_{30-90N}$ (Table 1) would have been surpassed by Pinatubo in 1991 and El Chichón (Mexico) in 1982 (Jungclaus

et al., 2017), but by none of the other 20[th] century eruptions.





## 3 Sources of mid-17[th] century eruptions

Ice core sulfur signals do not allow discrimination between potential volcanic sources. Attribution only becomes possible if written documents confirm the coincident eruption of a given volcano (Lavigne et al., 2013; Verbeer, 1884) or (more securely) where tephra conserved in ice cores bears the geochemical fingerprints of a specific volcanic source (e.g., Jensen et al., 2014; McConnell et al., 2020). For none of the mid-17[th] century eruptions has tephra been analyzed in the ice cores so far, such that attribution of sources to the 1637, 1641-42 and 1646 sulfur spikes remains provisional. The 1637 sulfur spike in Greenland ice cores can potentially be ascribed to an eruption of **Mount Hekla** (63°N, Iceland) which erupted at VEI 3 on May 8, 1636. The eruption lasted more than a year and the estimated $5\times10^7$ m$^3$ of ejected tephra damaged pasture in north-eastern Iceland where it killed livestock (Thorarinsson, 1970). A spatially heterogenous sulfate distribution (note the absence of sulfate enrichment in the Summit15 ice core) is typical for eruptions in close proximity or upwind of the ice core sites. Sulfate deposition from distant eruptions with transport primarily occurring in the stratosphere is characterized by more uniform sulfate deposition over the ice sheets (Gao et al., 2007), as illustrated by the Veiðivötn 1477 eruption from Iceland (Abbott et al., 2021). The 1646 sulfur spike is often ascribed to **Shiveluch volcano** (56°N, Kamchatka Peninsula), which produced substantial pumice fall, voluminous ignimbrites and pyroclastic flows in the mid-17[th] century. As the first Russian Cossacks arrived in Kamchatka only by the end of the 17[th] century, the eruption cannot be found in historical sources. Multiple trees were, however, killed by the pyroclastic flows and preserved in volcanic deposits. As the bark and some growth rings were burned by these flows, kill dates range between 1646 and 1649, with the latter likely corresponding to the absolute date (Solomina et al., 2008; Brown et al., 2014). The volcanic eruption(s) in 1640 and 1641 have been described in much greater detail and ascribed to Koma-ga-take (Japan) and Mount Parker (Philippines). Both eruptions are discussed below.

### 3.1 Koma-ga-take

Koma-ga-take (駒ヶ岳; 42°N, 140°E; 1131 m asl), also known as Hokkaidō Koma-ga-take (北海道駒ヶ岳), Oshima Koma-ga-take (渡島駒ヶ岳) or Oshima Fuji (渡島富士), is an andesitic stratovolcano located on Hokkaidō, Japan. According to the Global Volcanism Program (GVP, 2013a), much of the truncated volcano is of Pleistocene age and the breached crater (resulting from an edifice collapse in 1640) remains visible (Yoshimoto and Ui, 1998). The collapse was presumably caused by a phreatic eruption (Fig. 3a).

The yearly records of Matsumae, or *Matsumae Nennen-Ki*, report that on July 31, 1640, part of Koma-ga-take's summit collapsed to send pyroclastic flows and a huge debris avalanche (estimated at 0.25 km$^3$) down its eastern slopes to produce a tsunami in Uchiura Bay (Katsui et al., 1975; Katsui and Chimoto, 1985; Yoshimoto et al., 2003), drowning at least 700 people in the present-day Toya and Usu region (Tsuji et al., 1994; Nishimura and Miyaji, 1995). Furious eruptive activity continued on August 1–2, 1640, so that contemporary sources report that "*the day became dark like nighttime with little sunshine*" (Katsui et al., 1975). Whereas ashfall continued, the color of the cloud turned purple on August 3 in Matsumae. Activity weakened substantially thereafter and stopped after about 70 days (Katsui and Ishikawa, 1981). Ashfall deposits of the Koma-ga-take's





1640 eruption are 2 m thick next to the volcano and still reach 10 cm in Tsugaru region (northern Honshu), some 130 km
south-southwest of the volcano (Yamada, 1958) for an estimated volume of 3.5 km³ ejected (Sasaki et al., 1970, Katsui et al.,
1986). The 1640 Koma-ga-take eruption is thus one of the largest in Japan during historical time, and deposited ash even in
central Honshu, located >800 km to the south.

## 3.2 Mount Parker

Mount Parker (6°N, 125°E; 1824 m asl), locally known as *Mélébingóy*, is a densely vegetated, andesitic-dacitic stratovolcano
near the southern tip of Mindanao Island (Philippines; GVP, 2013b). Its English name originates in 1934 when American
General Frank Parker spotted the mountain and claimed its discovery during a flight he piloted (Davis, 1998). Mount Parker
has a base diameter of 40 km; its summit is truncated by a 2.9-km wide caldera with steep walls rising 200-500 m above Lake
Maughan (*Lake Hólón* locally; Fig 3b).

The 1641 eruption was first described in western sources by the French naturalist *Guillaume Le Gentil* who visited the region
some 140 years after the cataclysm. In his books "*A Voyage in the Indian Ocean"* (published in 1779 and 1781), he erroneously
referred to a big eruption in January 1640 [sic] that would have devastated large parts of the Philippines. Relying on
contemporary Hispanic sources and geological investigations, Delfin et al. (1997) provided more insights into the sequence of
the eruption. Mount Parker started erupting on 26 December 1640, and produced a major explosive event on 4 January 1641,
resulting in the formation of the caldera as well as voluminous pyroclastic flows and lahars on its northern and eastern flanks
(Delfin et al., 1997). The eruption produced pumice comparable in type to those described for the 1991 Pinatubo eruption
(David et al., 1996).

While this eruption has been known from contemporary Hispanic documents, an eyewitness report of Jesuit monk Raymundo
Magisa written in 1641 has long been misattributed to Gunung Awu volcano in Indonesia (3°N, 125°E), *c.* 300 km south of
Mount Parker. Dating of 9 charcoal pieces and oral traditions of local indigenous people finally allowed attribution of the 1641
eruption described by Magisa to Mount Parker (Delfin et al., 1997). The same authors also translated Magisa's text testifying
to the eruption: *"On (…) 4 January by 9 AM, the (supposed) artillery fire increased to such an extent that it was feared that
the squadron (sent to investigate) might have run into some Dutch galleons. It lasted for about half an hour. People soon
became convinced, however, that the noise originated from a volcano which had opened up, because by noon we saw a great
darkness approaching from the south (…) By 1 PM we found ourselves in total night and at 2 PM in such profound darkness
that we could not see our hands before our eyes. This darkness (…) lasted until 2 AM [5 January 1641] when a little moonlight
appeared, to the great relief of Spaniards and Indios who had feared that they would be buried beneath the mass of ash which
had started to fall on them at 2 PM. (…) At 10 AM the ships found themselves in such deep darkness and terrible blackness
that they believed the Judgement Day to be at hand. It started raining so much rock, soil, and ash that the ships considered
themselves in grave danger and had to light lanterns and quickly lighten the heavy load of soil and ash that was accumulating"*
(Delfin et al., 1997, p. 35).



## 4 Tree-ring-based summer temperature reconstruction for the mid-17<sup>th</sup> century

The climatic conditions prevailing during the mid-17th century are quantified with a network of NH and European tree-ring records and complemented by excerpts from the vast corpus of historical records referring to harsh and hostile conditions in the late 1630s and early 1640s.

### 4.1 Northern hemisphere summer temperatures

To quantify NH summer temperature (JJA) and identify possible volcanic cooling induced by mid-17th century eruptions, we employed the *NVOLC* v2 reconstruction of Guillet et al. (2017, 2020). *NVOLC* v2 comprises 25 proxy records (13 tree-ring width [TRW] and 12 maximum latewood density [MXD] chronologies). A principal component regression (PCR) approach was applied to reconstruct NH JJA temperature anomalies (compared to the 1961–1990 mean). A bootstrap method was then combined with the PCR to estimate the skill of the reconstruction and to compute confidence intervals of reconstructed JJA temperatures. To gradually adjust to the changing number of records available back in time, the PCR was combined with a nested approach (Guillet et al., 2017). *NVOLC* v2 reconstruction is consequently based on 23 subsets of tree-ring chronologies or nests, with the most replicated nest containing 25 chronologies spanning the period 1230–1972. For each nest, a principal component analysis was calculated on the proxy predictors and the first *n* principal components (PCs) with eigenvalues >1 were retained as predictors to develop a multiple linear regression model. The regression models were all calibrated on JJA temperature obtained from the Berkeley Earth Surface Temperature (BEST; Cowtan et al., 2019) dataset over the period 1805–1972. For each nested regression model, a split calibration–verification procedure was repeated 1000 times using the bootstrap method to assess the robustness of the transfer function and 1000 reconstructions were generated. The final reconstruction of each nest is the median of the 1000 realizations and is given with its 95% bootstrap confidence interval. To quantify the reconstructed cooling within a context of climate variability at the time of the major eruptions under study, we filtered the *NVOLC* v2 series with a 31-yr running mean (*NVOLC_filt* v2). For example, in the case of the year 1640, a background was calculated by averaging the window 1625–1639 and 1641–1655. The magnitude of the anomaly is then determined by subtracting this background from the temperature reconstructed in 1640.

Figure 4 illustrates the cooling induced by major volcanic eruptions of the 17th century. The reconstruction yields cooling of varying magnitude after some of the major eruptions of the 17th century: According to the NH JJA *NVOLC* v2 temperature reconstruction, the largest cooling induced by any eruption of the past 1500 years occurred after 1600 Huaynaputina in Peru (White et al., 2021) with NH temperature reduction of –1.56 °C in 1601 relative to the 1961–1990 reference period, followed by –1.48°C in 1816 (rank 2, Tambora), –1.42°C in 1699 (rank 3, absent apparent volcanic activity), –1.40°C in 536 (rank 4, unidentified source), –1.35°C in 1453 (rank 5, unidentified source; Abbott et al., 2021) and the notable eruption cluster between 1108 and 1110 (–1.33°C in 1109, rank 6, presumably Mount Asama, among others; Guillet et al., 2020). The assumedly largest volcanic eruption of the Common Era occurred in 1257 (Samalas; Lavigne et al., 2013; Guillet et al., 2017) and induced a cooling of –1.04°C in 1259, ranking 16th in the *NVOLC* v2 reconstruction.





The years 1641 and 1643 stand out as the thirteenth and seventeenth coldest years of the last 1500 years with –1.13 and –1.02°C anomalies, respectively. By contrast, cooling was less marked after the assumed 1636 Hekla (–0.15°C; rank 628) and the 1646 Shiveluch (–0.36°C; rank 348) eruptions.

### 4.2 Reconstructed regional variability of summer cooling between 1638 and 1643

To estimate regional variability of summer cooling induced by the mid-17th century eruptions, we used the climate field reconstruction of extratropical NH JJA temperatures developed by Guillet et al. (2017, 2020) spanning the last 1500 years. The target field (predictand) used for the reconstruction is the BEST JJA gridded temperature dataset (1° × 1° lat-long grid). The NH was divided into 11 regions based on the spatial distribution of the 25 chronologies and their correlations. Performance and temporal stability of reconstruction models was assessed for each grid point with a split-sample calibration and verification approach repeated 1000 times using the bootstrap method.

Figure 5 shows JJA-gridded reconstructions for some of the largest CE eruptions (Fig. 5a) and all years between 1638 and 1643 (Fig. 5b). As with other eruptions, we observe widespread, yet heterogeneous NH summer cooling, above all in 1641. The most affected regions are the Western U.S. (up to –2°C) Scandinavia (–1.5°C), Siberia (–1°C) and Central Asia (–1°C) and, to a lesser extent, Western Europe (–0.4°C). Whereas Central Asia, Scandinavia and the Western U.S. experienced cooling until 1643, summer temperatures recovered more rapidly in the other regions.

### 4.3 European summer temperatures between 1638 and 1643

The European summer temperature reconstruction uses twelve multi-centennial TRW (i.e., Forfjorddalen, French Alps, Tatra Mts), MXD (Jamtland, Lauenen, Lötschental, Nscan, Pyrenees, Torneträsk, Tyrol) and stable oxygen isotope (Angouleme, Fontainebleau) chronologies covering Western Europe and Scandinavia.

A principal component regression (PCR) approach (Stoffel et al., 2015; Guillet et al., 2017) was applied to reconstruct summer temperature anomalies from these chronologies. Regression models were calibrated on JJA temperatures obtained from the E-OBS pre-1950 and v22.0e gridded datasets (0.25 × 0.25° lat/long) over the period 1920–2010. For each regression model, a split calibration–verification procedure was repeated 1000 times using a bootstrap method to assess the robustness of the transfer function and to generate 1000 reconstructions. The final reconstructions for each grid point comprised the median of 1000 realizations and covered the period 1500–2010. For each reconstruction, we calculated coefficient of determination ($R^2$ for the calibration period) and squared correlation coefficient ($r^2$ for the verification period), RE (reduction of error) and CE (coefficient of efficiency) statistics. Anomalies are given with respect to a moving 31-yr average calculated around the years given in Fig. 6.

The gridded summer temperature reconstructions allow analyses of warm and cold spells that have affected Europe. Although not comparable to the cooling observed in Europe after the 1600 Huaynaputina (White et al., 2021; not shown) eruption (with –1.87°C in 1601; rank 1), 1641 stands among the coldest years of the last 500 years (–1.19°C; rank 16), with a cooling comparable to that observed after the Tambora eruption (–1.23°C in 1816; rank 14). Notably, cooling is most obvious in



Scandinavia where temperatures were more than –2°C below average (compared to the 31-yr moving average around the
event), but much less marked in the Alpine and Mediterranean domains. The cooling becomes much more generalized across
most of Europe in 1642 (–0.47°C; rank 105) and in 1643 (–1.01°C; rank 27).

While it is tempting to attribute the strong, yet localized cooling observed in Scandinavia in 1641 as well as the more
widespread, but less marked, cooling in 1642 and 1643 to the eruptions of Koma-ga-take and Mount Parker in July 1640 and
January 1641, at least some cooling may be the result of natural climate variability. Indeed, cold spells like those observed in
the early 1640s and affecting most of Europe were frequent in the 17th century and occasionally occurred absent any known
major volcanic activity, for instance in 1608, 1609 or 1633 (–1.57°C; rank 4).

## 5 Evidence for dust veils and volcanic aerosols over Europe in 1641 and 1642

In July 1641, several contemporary texts report "thick fog" in different regions of Germany. The Landgrave Hermann IV of
Hesse (1607-1658) observed the weather around *Rotenburg an der Fulda* (Hesse) and in other locations on his frequent travels.
On several occasions, he describes the almost continuous occurrence of fog after July 17, 1641 (Historical source *HS1;
appendix*). The "thick fog" returns on July 19–20 and persists from August 1 to 4 (*HS2*). The weather is often described as
"dull", sometimes also reasonably pleasant and warm, *"but not quite bright"* (August 7, 1641; *HS3*). A "thin fog" is described
on multiple other occasions in August and early September, 1641. On August 13 and 15, the fog was so dense that the weather
could not be observed (*HS4*). Interestingly, on August 21, a *"nicely warm"* day was followed by *"fog and hoar frost"* after
midnight (*HS5*), a phenomenon observed again on August 23, 1641. On September 6–7, weather observer Hermann IV of
Hesse described the weather as "dull and smoky" and used the word "*heerrauchig*" – a term that was often used in German to
describe volcanic haze, even if observers were historically unaware of the origin of such dust veils (*HS6*, Demarée, 2014;
Kleemann, 2019). From the evening of September 7, 1641, he continued to make observations from the town of *Eisenach
(Thuringia)*, where he witnessed fog on September 7–8 and 12 as well as hoar frost on September 8–9 (*HS7*). On September
18, 20 and 22, the same observer noticed fog in *Kassel* and in *Rotenburg an der Fulda* (*HS8*). Fog was also observed in *Paris*,
where Canon Jean de Toulouse of the Abbey of Saint-Victor stated that "*from August 25, 1641, cold rains arrived,
accompanied by unusual fogs and caused many diseases and a high mortality in the city until March 1642*".

Hermann IV of Hesse's multiple notices of *heerrauch* are particularly noteworthy. These are found in March 1641, recurring
in April, May and July 1642. But we may question whether all observations are linked to a volcanic dust veil or should be
ascribed to forest or turf fires. During summer 1641, however, it seems unlikely that forest or turf fires made a significant
contribution to the fog and haze as the season had been described as wet and cold, with conditions rather unfavorable for
ignition.

Independent evidence for the presence of volcanic aerosols in the stratosphere comes from a dark lunar eclipse at 01:52 UT on
April 25, 1642. The luminosity and brightness of total lunar eclipses are a reliable indicator of stratospheric turbidity (Stothers
2004, 2005; Guillet et al., 2020). While copper-red lunar eclipses generally point to a clear stratosphere, the eclipsed Moon





tends to appear darker or even to disappear entirely if the stratosphere contains sufficient volcanic aerosols. Dark total lunar eclipses were thus observed following the 1257 Samalas, the 1600 Huaynaputina, the 1884 Krakatau, the 1912 Katmai-Novarupta, the 1963 Agung, the 1982 El Chichón and the 1991 Pinatubo eruptions (Guillet et al., 2020). For our period, the astronomer Johannes Helvetius (1611-1687) reports, in his *Selenographia* published in 1647, that the location of the moon

could not be ascertained during the lunar eclipse of April 25, 1642, even with the aid of the telescope, although the air was sufficiently pure to discern the stars (Helvetius, 1647; *HS9*). This account contrasts starkly with usual records of total lunar eclipses depicting blood-red eclipsed moons and suggests that a significant volume of volcanic aerosols still remained in the stratosphere in April 1642, i.e., 15 months after the eruption of Mount Parker.

## 6. Weather anomalies and societal distress in the late 1630s and early 1640s

Weather anomalies were widespread in the late 1630s and the early 1640s, as described in diverse contemporary sources. The following subsections present evidence for unseasonal, sometimes even severe, weather conditions in Western, Central, and Northern Europe as well as in China and Japan. Given the vast number of historical documents surviving from the mid-17[th] century, we can provide only a glimpse of what exists regarding weather data at the time of the Koma-ga-take and Mount Parker eruptions, and in the years following.

**6.1 Western and Central Europe**

Western Europe of the early to mid-17[th] century was dominated by the Holy Roman Empire, which roughly included the territory of modern-day Austria, Belgium, Czechia, Denmark, Germany, northern Italy, Luxemburg, westernmost Poland, the Netherlands, Slovenia, and Switzerland. The Empire was both religiously and politically fragmented and suffering from the Thirty Years' War (1618–1648), with conflict occurring predominantly within its territory. The war involved several European

powers and fell along the different religious denominations – Catholicism and Protestantism. The spread of diseases, the rise of poverty, lack of resources, quartering of soldiers, general violence and looting also cost many lives and created immeasurable hardships for populations within the Holy Roman Empire over three decades. In 1620, the Holy Roman Empire had about 16 million inhabitants; around 1650 it had only 10 million (Gotthard, 2016). In the early 1640s, the war was in its final phase, and suffering from destruction, disease and hunger was likely enhanced by unseasonal weather in the early 1640s.

Contemporary chroniclers detail harsh weather in the early 1640s and numerous eyewitness reports underline the cold and wet conditions prevailing across the region. Both **spring and summer 1640** are reported as being wet and cold, with many rivers flooding. For the territory of modern-day *Switzerland*, March temperatures sank to such a low that smaller water bodies were covered with ice and snow on the Swiss Plateau into April (*HS10*). Grain harvest on the Swiss Plateau was also poor (Pfister, 1984). The following **winter 1640/41** was long and severe, followed by a rather wet and cool spring. In the area of present

Switzerland, the winter 1640/41 saw heavy snow. On the Swiss plateau, snowfall occurred in late March and April (*HS11*), followed by a late frost in May (*HS12*). **Summer 1641** was cold across Europe, ice formed, for instance, several times in



southern Germany [Middle Franconia], and especially on July 25, 1641 [August 4, 1641] (*HS13, 14*), and summer was reported as anything but sultry in Europe even during dog days. In Krupka (today's Czechia), frost was noted on August 10, 1641 — and cucumbers and cabbages froze (Brázdil et al., 2004).

At *Frankfurt an der Oder*, harvest was affected by incessantly rainy, cold weather, and thunderstorms (*HS15*), and the frost in **mid-September 1641** destroyed wine in southern Germany (*HS16*): *"The wine of 1641 was sour […], grapes did not grow at all during the dog days, [and] before harvest they froze on the vines"* (*HS17*). Various independent sources confirm the presence of strong hoar frost around mid-September 1641 in Germany. In the area of present *Switzerland*, the harvests were poor and late as well (Pfister, 1984).

In **May 1642**, much of present-day Germany saw a cold spell: In *Bavaria*, *"the coldest winds and hoar frost"* continued until mid-May, damaging grain and fruit trees greatly (*HS18-19*). Late snowfall and hard frosts are reported for May 1642 in *Krupka*, causing damage to the vineyards (Brázdil et al., 2004). Hoar frost and snow are again reported for several days in southwestern Germany (*HS20*) and Krupka (Brázdil et al., 2004) in June 1642. By contrast, a report from Diez (Lahn) in today's *Rhineland-Palatinate* noted that the **summer 1642** was calm and the people in Diez harvested exceptionally good crops (*HS21*).

The *Baden* wine chronicles allow a comparison of the quality of the wines produced each year during the 17th century. The 1630s and 1640s clearly appear to be very harmful for wines. Of the 1630s, only the years 1631 and 1638 produced "good" wine, while the others produced *"bad"* or *"sour"* wine. Between 1640 and 1650, just four years (i.e., 1644–1647) provided *"good"* wine. For **1641**, the Baden chroniclers indicate that the vine suffered late frost in May and then early frost in September (Garnier, 2010).

In neighboring France (not part of the Holy Roman Empire), the grape harvest between 1640 and 1643 began a full month later than usual and grain prices surged, indicating poor cereal harvests, especially also in eastern France. A source from *Dijon* described **summer 1640** as chilly, resulting in a grape harvest that was comparably late. Poor temperatures also resulted in high grain prices on *Île de France*. Sixteen forty-one saw considerable precipitation during winter and rather low summer temperatures. In 1642, the grape harvest was again late in Dijon and very late around Paris, while grain prices increased rapidly
in the Île de France, probably the result of cold conditions prevailing in 1642 (Parker, 2013).

Grape harvest dates in *Besançon* (France) are intermittent in the 1630s and 1640s, only available for 1631, 1633, 1635, 1638-1642, 1646 and 1649. In these years, harvests occurred late between October 1 and 15 (Garnier et al., 2010). Years without harvest dates are not necessarily explained by meteorological factors. Most correspond with the arrival of the Swedish and French armies in Franche-Comté (1632, 1641, 1644, 1639, 1644), or can be related to the quartering of Swiss mercenaries
(1634), damage to vines by 'German soldiers' of the Duke of Lorraine (1637, 1647) as well as plague (1630 and 1635). Only three years without harvest (1636, 1648 and 1650) are attributed directly to meteorological conditions by the city clerks: more exactly to frequent rainfall episodes between June and October. Abnormally rainy conditions were obviously sufficient to justify *"pro serenitate"* processions during which people prayed for cessation of the rain (Garnier, 2010).

*Paris* has excellent climate observations between 1630 and 1640 in municipal acts. Parisian aldermen ('échevins') met almost
weekly to deliberate on city affairs. Among them, climatic events, more particularly extreme events, were systematically



discussed. This is easy to understand since droughts, floods or even freezing of the Seine had an almost immediate impact (in one or two days) on prices in Parisian markets. Indeed, until the mid-19th century and the rise of the railroad, heavy goods such as wheat and wood used by city dwellers for heating and cooking were mainly transported by waterways (Garnier, 2015). The municipal acts also provide other indicators for droughts, mentioning the drying up of the sources and public fountains

supplying the main hospital and conditions that prevented hydraulic mills from making flour. Between 1630 and 1650, Paris thus suffered five very severe droughts (some long like those of 1631-1632 and 1646-1647), but the drought of 1639 was undoubtedly the most dramatic of the 17th century (Garnier, 2019). Between 1630 and 1650, freezing of the Seine affected Paris six times (1635, 1636, 1637, 1644, 1645 and 1646). In the latter case, it is likely that the freezing was favored by the drought which preceded it in summer and the autumn, so that when cold arrived, low water levels facilitated ice formation that

lasted nearly three weeks.

**6.2 Northern Europe**

In the 17th century, most of modern-day Sweden, Finland, Estonia, and northern Latvia belonged to the Swedish Realm, a 'great power' of its time. Like in the Holy Roman Empire, contemporary documents point to severe weather and resulting agricultural hardship in the early 1640s. *Estonian* and *Livonian* sources report unusual heat and drought in **summer 1640** to a

degree that grain crops were damaged (Tarand et al., 2013), whereas the following two summers were cold and rainy (Soom, 1940). Grain ripening was badly delayed and the first autumn frosts in early September 1641 destroyed a great part of the harvest. **Winters 1640–41** and **1641–42** were notably cold (Tarand et al., 2001). Crop failures with consequent food shortage troubled Estonia in 1641 and 1642 but did not likely escalate into an acute subsistence crisis (Seppel, 2008). In the territory of present-day *Sweden*, bad crop failure was reported in 1641, especially in northern Sweden (Leijonhufvud, 2001). In Stockholm,

at the heart of the realm, grain prices were 13 to 20% higher in 1641–1643 than the 1630s average (Edvinsson and Söderberg, 2010). Southern Sweden suffered a "*violent fever or burning disease epidemic* [likely typhoid fever and typhus which are hunger-related diseases], *which killed both rich and poor*" especially in late spring 1642 (*HS22*).

Within the Swedish Realm, the area of modern-day *Finland* was likely the most affected by the weather anomalies of the 1640s. Already in 1632–1635 the region suffered from local poor harvests, and the year 1633 witnessed a country-wide crop

failure (Johansson, 1924). In **August 1638,** a week-long frost killed the crops and allegedly "hand-deep" snow covered the ground (Johansson, 1924). Like in *Estonia*, cold conditions in **summer 1641** delayed the ripening of grains and frosts in early September killed the grain (Mårtensson, 1952). Grain tithes from western Finland indicate that the 1641 harvest was half of normal (Huhtamaa and Helama, 2017); harvests remained poor in the following year, with yields only reaching approximately two thirds of the mean (Huhtamaa and Helama, 2017). Unlike in Sweden and Estonia, the crop failure in 1641 triggered severe

hunger in Finland. The economic consequences of these harvest losses can be seen in many regions, with every fifth or fourth farmstead marked as 'deserted' in the administrative records (Huhtamaa et al., 2021). The high percentage and spatial extent of abandoned and acutely impoverished farmsteads in the mid-1640s point to a severe agricultural crisis, similar to the early 1600s when the eruption of Huaynaputina in 1600 (White et al., 2021) triggered harvest failures, famine and farm desertion





(Huhtamaa et al., 2021). Whereas in the Swedish Realm the year 1640 was hot and dry, unusual cold and constant rain ruined

the hay in *Iceland* in **1640**, such that farmers resorted to dried fish as cattle fodder (Parker, 2013).

In *Ireland*, **winter 1641-42** was particularly severe. Numerous contemporary documents describe the severity of cold. Particularly important are the approximately 8,000 surviving statements of mainly Protestant interviewees (or "deponents") recounting their experiences during the 1641 Rebellion, as attested to British government authorities, beginning with the work of the Commission for Distressed Subjects in December 1641. In these documents we hear survivors (often widows from

Ulster and Leinster) describing *"cold snowy weather"* and *"the extremity of the winter"* (*HS23*). This unusual cold exacerbated the consequences of the rebelling Gaelic and "Old English" Catholics, who destroyed or turned Protestant residents out of their residences, in addition to their stripping and forced expulsion (Fig. 7). Thus, Katherin Cooke of Armagh recalled *"such frost [and] snow that the deponent[']s children and divers [many] other children [...] present at that battle would[,] where they saw the warm blood of any fall on the ground[,] thread therein with their bare feet to keep them from freezing and starving*

*such was the extremity of the weather"* (*HS24*). Another deponent recalled: *"Whereupon the deponent and his wife and 5 small children going away were stripped of all their clothes [...] one poor daughter of his seeing him [...] grieve for their general miseries in way of comforting said she was not cold nor would [she] cry although presently after [...] she died by that cold and want: and the first night this deponent and his wife creeping for shelter into a poor crate were glad to lie upon their children to keep in them heat and save them alive"* (*HS25*). Other sources also provide graphic illustrations (often with

propagandistic intent), such as that from James Cranford's, *The Teares of Ireland* (1642; Fig. 7), the caption from which reads: *"English Protestantes striped naked and turned into the mountaines in the frost, and snowe, whereof many hundreds are perished to death, and many liynge dead in diches and savages [mainly Irish Catholics] upbraided them saynge now are ye wilde Irish as well as wee".* These conditions have, to date, been associated with explosive volcanism by Ludlow et al. (2013) and more specifically the eruption cluster starting in 1636 by Ludlow and Crampsie (2018, 2019).

**6.3 China and Japan**

The *Compendium of Chinese Meteorological Records of the Last 3000 Years* (hereafter the *Compendium*; Zhang, 2004) is the key reference for past weather and climate anomalies for the territory of present-day China. The *Compendium* is compiled from 7930 volumes of the state annals and local chronicles, supplemented by representative personal diaries. For the Ming and Qing Dynasties (1368–1911), records can be considered precise at the county level and reported place names have been linked

to the corresponding modern county. Here, we adapt a simple sorting and counting approach to digitize documentary records (Gao et al., 2017) and to quantify the occurrence of droughts or other disastrous events in each county (Xiao et al. 2015; Gao et al., 2021a).

At the time of the 1640s eruptions, China was suffering one of the most disastrous droughts in its known history, called the *Chongzhen Great Drought* (Chen et al., 2020). Extraordinarily dry conditions (Fig. 8) resulted in severe societal and economic

consequences during the final years of the Ming Dynasty (1638–1644), perhaps ultimately contributing to its collapse. Drought started to emerge in the provinces of Shandong, Henan, and Shaanxi along the Yellow River Basin in **1638**, with 91 counties





reporting drought. Reports rose to 109 counties in **1639** and 303 in **1640**, almost six times the average annual drought count during the 17[th] century. Indeed, 1640 may have been the single driest year in the region since at least 1470 (Zhang and Lin, 1992). The drought expanded from the Yellow and Huai River valleys to the middle and low reaches of the Yangtze River in

1641 (Fig. 9). Local sources report that from the Huai Valley to the Northern Metropolitan Region, *"all the bark had been stripped from the trees and people [even] dug up corpses for food"* (Zhang et al., 1974). In the early months of 1641, unusually heavy snowfall was then recorded in southeastern China, while conditions remained very dry with infestations of locusts in Northern China from where they propagated rapidly to the Yangtze River Valley to form a large-scale locust plague (Song, 1992; Liu et al., 2018). The drought and locust plague resulted in an estimated reduction of 20-50% in per capita grain

production (Zheng et al., 2014), triggering a sharp increase in rice prices that intensified widespread famine (Fig. 8).

The severity of the drought was such that part of the Grand Canal in Shandong Province dried out completely, an incident without documented parallels over the past 3000 years. Water flow of the Yellow and Yangtze Rivers as well as lake levels in the Zhejiang, Shaanxi, Shandong, Henan, Shanxi, and Hebei Provinces dropped and eventually dried up in **1640–1641** (Zhang, 2004). Shen et al. (2007) suggested that the intensity of the *Chongzhen Great Drought* (1638–44) was comparable to that of

1997, in which about 700 km of the lower Yellow River dried up, stopping outflow to the Yellow Sea for more than 300 days. The *Chongzhen Great Drought* was likely the outcome of natural climate variability intensified by the Mount Parker eruption. Chen et al. (2020) ascribe this intensification to a decreased land-sea thermal contrast and a negative soil moisture response combined with a weakening and eastward retreat of the West Pacific Subtropical High.

In many provinces, purported incidents of cannibalism occurred, and a widespread epidemic was likely triggered or

compounded by malnutrition and weakened immune systems. By **1641**, 29 counties within Shandong Province reported this pestilence. Conditions are described in the *Gazettes of Tangye County*: "*No rain again in spring, a bucket [a unit used to measure grain, about 7.5 kg in the Ming Dynasty] of rice worth more than a thousand QIAN [a unit used to measure money, about 3.125 g in copper], cannibalism among family members, corpses buried for weeks were dug out for food. Pestilence raged in summer, no household escaped; [...] eight to nine out of ten people died of pestilence, hunger and robbery [...] it was*

*indeed a situation that has never been seen or heard of"* (Zhang, 2004, v. 2, p. 1612).

Between **1641 and 1646**, rebellion and warfare reached historical highs (Zhang et al., 2006; Zheng et al., 2014). The year **1645** ranks first in war frequency for the period 850–1911 (Fig. 8), with 16 wars in central China and 6 rebellion wars elsewhere in the country. With parallels to Ireland (Ludlow and Crampsie, 2019), the counties in which uprisings occurred were most frequently those where extreme weather was most severe, suggesting that regional drought severity played a critical role in

fueling peasant uprisings (Zheng et al., 2014).

*Japan* was likewise affected by a famine already underway at the time of the Koma-ga-take and Mount Parker eruptions in 1640 and 1641. The so-called *Great Kan'ei Famine* (寛永の大飢饉 or Kan'ei kikin) started in 1640 and lasted until 1643, with up to 100,000 people and innumerous livestock dying from starvation and disease (Yamaguchi and Sasaki, 1971; Asao, 1975; Nagakura, 1982). The Kan'ei kikin was the first major famine to occur in Japan after the rapid urban and demographic growth

of the late-16[th] and early-17[th] centuries that made many more Japanese dependent upon others for their food supply (Atwell,



1986). The eruption of Koma-ga-take on 31 July 1640 must thus be ascribed a compounding role to an already imminent crisis: heavy ashfall resulted in plant poisoning near the volcano and triggered local crop failure over two years. In **1641**, the situation worsened as incessant rainfall and unusual cold propelled another poor harvest in the northeast (Yamamoto, 1989; Atwell, 2001). **Winter 1641-42** was cold as well and **summer 1642** was cool, diminishing harvests once again in northeastern Japan,

whereas floods and droughts destroyed harvests elsewhere in the country (Atwell, 2001).

In **1642**, *"the corpses of those who had starved to death filled the streets while the peasants, artisans, and merchants who begged for food were numerous. None of these people had any clothes [and could only] wrap straw and other matting around their bodies for protection"* (Kanei nikki zōho in Asao, 1975). By **June 1642**, starving peasants started to abandon their lots to the degree that the governor ordered the abolition of tobacco plantations and strongly restricted alcohol production to favor

food crops. Regardless, death tolls increased in **winter 1642-1643** such that farmers performed infanticide of children below 7 years and lent older children, often to pimps, in Fukushima Prefecture. The famine ceased only later in 1643 when crops yield rebounded.

## 7 Climate, volcanic eruptions and societal crisis: correlations exist, but is there also causality?

Volcanic eruptions have been demonstrated to influence climate by ejecting gases and solid particles into the atmosphere

(Robock, 2000; Timmreck, 2012; Sigl et al., 2015), where sulfur-containing gases are converted into aerosol particles (Zanchettin et al., 2012). These volcanic aerosol particles influence Earth's radiative balance by (i) scattering incoming solar radiation back to space, a process that will cool the troposphere and the surface, and by (ii) absorbing long-wave solar and terrestrial radiation, leading to a local stratospheric warming (Fischer et al., 2007; Christiansen, 2008; DallaSanta et al., 2019). At the same time, volcanic eruptions are also known to affect global precipitation patterns (Iles et al., 2013; Rao et al., 2017;

Zuo et al., 2019), including a weakening of the African (Khodri et al., 2017; Manning et al., 2017; Zambri et al., 2017) and/or the East Asian monsoon (Zambri et al., 2017; Dogar and Sato, 2019).

### 7.1 Worsening climatic conditions and advancing glaciers in the European Alps

The early decades of the 17[th] century mark the start of prolonged limited sunspot activity known as the "Maunder Minimum" (1645–1715). This period also marks a phase during which climatic conditions were worsening, with colder conditions also

translating into advancing glaciers. In the Alps, contemporary sources report the disappearance of fields, farmsteads and entire villages as various glaciers reached their furthest extent in historical times (Nussbaumer and Zumbühl, 2012): The Mer de Glace (Mont Blanc, France), for instance, had one of its Little Ice Age maxima in 1644 after advancing by more than 200 m between 1641 and 1644. This points to a sustained, positive mass balance of the glacier, implying a multi-year succession of cold and wet conditions. By contrast, the Lower Grindelwald Glacier (Switzerland) reached its Little Ice Age maximum

somewhere between 1600-1606 CE and 1636-1645 CE, with a second peak in 1855/6. The Upper Grindelwald Glacier had its





largest extension between 1593 and 1630 CE. Indeed, various glaciers across the European Alps reached valley floors at the beginning or during the first decades of the 17th century (Sigl et al., 2018).

In 1642, officials of the village of Chamonix (France) spoke of the daily progression of the glacier and evoked the avalanches of snow and ice which continued to carry away houses and livestock. The same was true for the Argentière glacier (France),

which threatened to consume the church of the same name. The Glacier des Bois (France)exhited the most impressive advance, and in 1644 threatened to block and transform the Chamonix valley itself into a lake. More serious in terms of human vulnerability was the growing threat of glacier rupture, as demonstrated by the Bossons glacier (France), which in February 1643 swept away a third of the Les Bois hamlet. Alpine mountain communities coped as best they could. Believing glaciers to be evil, they invoked divine protection, at first resorting to rogations to implore God's help. Faced with the ineffectiveness

of this strategy, they then requested the intervention of the Bishop of Geneva who in 1644 led a procession there. Thereafter, the ice slowly receded (Le Roy Ladurie, 2004).

## 7.2 Political instability, religious disputes, warfare and abandonment of farms

The early decades of the 17th century were characterized by major warfare and widespread religious disputes. The most widely known is the *Thirty Years' War* (1618–1648), consisting of a series of battles fought primarily in present-day Germany,

resulting in one of the deadliest wars of Europe. While at first restricted to battles among Catholic and Protestant states of the *Holy Roman Empire*, fighting became part of a wider European struggle after 1635, with the Swedish Realm, Dutch Republic and France on the one side, and Spain and the Austrian Habsburgs (including large parts of the Holy Roman Empire) on the other. Estimates of military and civilian deaths range from 4.5 to 8 million (Outram, 2002; Parker, 2008), many, however, from disease, starvation, or violence against civilians. Some sources suggest mortality of up to 60% of the population in certain

areas of present-day Germany (Outram, 2002), while in the Palatinate and Württemberg, losses are estimated at between 70 and 90%, and in Alsace they average 60% (Outram, 2002). These deaths must be considered the result of combined severe weather conditions (induced or at least often compounded by volcanism), epidemics, military violence, and famines. It is difficult to determine which was the most decisive factor, particularly given their likely interdependencies, but the scale of the combined impact is less in doubt. Evidence suggests that mortal sickness in military camps and peasant communities far

exceeded death in action, perhaps in the proportion of five to one (Outram, 2002).

The entry of France into 'open war' in 1635 brought a considerable increase in taxation, contributing to revolts almost every year for 10 years, mainly in the south and southwest of France. The movements of troops served as a vector for the plague which spread widely. Famous for his hostility to d'Artagnan and his musketeers in the novel by Alexandre Dumas, Cardinal Richelieu is better known to historians for having created in 1638 the heaviest regular tax, in the form of the 'wintering place'

(*quartiers d'hiver*) for troops. This obliged subjects of the King of France to provide food and shelter for soldiers during winter. Requisitions for food and equipment such as mills, ovens and means of transport were also often fulfilled in a context of pillage and brutality. A further cause of discontent was the *'tailles'*, comprising all direct taxes. These, moreover, were paid almost entirely by the peasants, and the Thirty Years' War was thereby waged heavily at the expense of peasant income in France. In





1648, the share of 'tailles' in the revenue of the royal state approached 62%. Popular riots among peasants and modest city
dwellers thus appear a response to fiscal and military aggression of absolutist authorities, and the socioeconomic 'efficacy' of
shocks to agricultural production arising from inclement weather in this period and region must be understood as operating
against this background.

The multi-faceted associations among climate, warfare and famine are exemplified by the case of Finland, being part of the
*Swedish Realm* in the early 17th century. The Swedish Realm intervened in the Thirty Years' War in 1630 CE and from the
late 1630s its army consisted increasingly of Swedish and Finnish conscripts as infantry was raised by conscripting every tenth
man among the adult male peasant population (Jespersen, 2016). The Realm's constant need for soldiers increasingly impacted
agricultural production, as call-ups were annual during wartime. In the case of Finland, conscriptions were particularly heavy
with at least 13,000 men recruited between 1638 and 1645 (Lappalainen, 1987). This number is considerable as the area held
only an estimated 27,500 peasant households at this time. Consequently, nearly every second farm was missing an adult male
from its agricultural workforce. Furthermore, the Swedish recruitment system created an additional financial burden, because
to avoid being shipped to the European battlefields, peasants started to hire men to take their place in the conscriptions
(Villstrand, 2000). Yet even if hired men, adult sons or farmhands were recruited preferentially, heads of farmsteads were
increasingly conscripted with time and as the number of available men decreased. By 1643, nearly 60% of all recruited men
from southern and central Finland were the masters of their farmsteads (Lång, 2003). The resulting labor shortfall undoubtedly
affected grain production quantity and quality. Even if battles were fought far from Finland, livelihoods were rendered more
vulnerable to environmental hazards, like the extreme cold of 1641. Thus, the interaction of warfare and extreme cold was key
to the early 1640s agricultural crisis in Finland (Huhtamaa et al. 2021), and it is arguably a matter of semantics (and/or
disciplinary norms) as to whether both are considered "causal" or "contextual" in this sense. The poor weather also diminished
yields in Estonia and the main agricultural areas in Sweden, but in being located further south than Finland, they were less
affected by the unseasonal summer weather conditions (Edvinson et al., 2009), while Estonia (and men from other newly
gained territories) also escaped the onerous conscription experienced by Finnish peasants.

In Ireland, one of the most infamous rebellions against English rule started in October 1641 with (mainly) Irish Catholics
trying to seize control of the English administration and killing many Scots and English Protestant settlers in what evolved
into the *Irish Confederate Wars* or Eleven Years' War (*Cogadh na hAon-déag mBliana* in Irish), continuing to 1653.
Remaining the most destructive conflict in Irish history to date, the 1641 Rebellion caused 200,000–600,000 casualties from
fighting as well as war-related exposure, famine and disease. The course of the initial 1641 rebellion is described in great detail
in documents known as the Depositions, comprising more than 8,000 (mainly Protestant) witness statements. These make
frequent reference to extreme cold during the 1641/42 winter (Section 6.2). Mapping these references per Irish county, while
accounting for the variable number of witnesses per county, reveals a pattern reassuringly consistent with meteorological
expectations, including less frequent references nearer warmer coastal and southern regions (Fig. 10). Comparing this
geography to that of reported killings (Smyth, 2013), while controlling again for variable witness numbers per county, reveals
a notable positive correlation (Spearman rho=0.596, p=0.003, n=32: Ludlow and Crampsie, 2019). One credible part-





explanation of this result is that regional variance in the severity of cold influenced the severity of violence, with Catholic populations that had become increasingly marginalized and vulnerable over the preceding decades of British rule more likely

to revolt where more severely affected by weather conditions.

This observation tells only part of the story, of course, though it is one largely absent from understandings of the Rebellion (its motivations and underlying contexts) until recently. The geography of violence during the Rebellion certainly reflected many additional political and economic factors, some more immediate to the time of the Rebellion and others longer evolving. Of the latter, mounting indebtedness among Catholic landholders in the decades preceding the Rebellion is important. To default

on a loan frequently meant the forfeiture of one's lands, and loans were thereby made by wealthy creditors seeking to increase their landholdings in Ireland during this pivotal era of expanding colonization ("plantation") by English and Scottish settlers (Ohlmeyer, 1998). Unsurprisingly, therefore, accounts from the Depositions outline how some Catholics sought to destroy loan agreements (bonds) amid the chaos of 1641/42 (Canny, 1995, 2001). But it is a mistake to consider such factors unrelated to the climate (and environment). While the environment has often been over-simplistically (e.g., mono-causally) attributed a

role in human affairs, its effective omission from many (perhaps most) histories similarly promotes more simplistic (or incomplete) understandings. Human societies cannot exist (and their histories did not occur) independent of the environment and its variability. Indeed, this inseparability or entanglement can be illustrated by returning to the 1641 Rebellion. Ludlow and Crampsie (2019) have thus shown a repeated association between harvest crises (of which there were many, frequently triggered by poor weather; Gillespie, 1989, 2018) and borrowing, used to cover related income shortfalls in the decades running

up to 1641 (Ohlmeyer, 1998). Extreme weather (and human responses to it) thus also played a role in the longer-term processes of mounting indebtedness, consequent loss of hereditary lands and growing grievance of Catholics towards Protestant settlers and elites that contributed to the Rebellion.

### 7.3 Widespread droughts and rinderpest in China and Japan

Adverse weather had already affected Chinese agricultural production in the years leading up to the early 1640s eruptions,

with some of the wealthiest regions suffering food shortage and disease by the late 1630s. This was at least partly the result of the *Chongzhen Great Drought*, featuring the five worst years of consecutive drought (1637-1641) in known Chinese history (Lamb, 1970; Atwell, 2001; Zheng et al., 2014; Chen et al., 2020). Food costs soared and lawlessness increased substantially. In April 1644, Beijing fell to a rebel army invading the capital from the economically devastated northwest (Liu et al., 2018). Only six weeks later the city fell again, this time to Manchu invaders, marking the start of a new era in Chinese political history

(Atwell, 1986).

Likewise, a succession of very cool summers in northern Japan and floods and droughts elsewhere in the country drastically reduced grain yields in the late 1630s and early 1640s, resulting in the *Great Kan'ei Famine* (Endō, 1982; Atwell, 1986). Notably, the famine occurred in a period when the center of Japanese rice cultivation was in the process of shifting from the south and west to the north and east, driven by the desire of lords in these regions to specialize in food production to exploit

growing urban populations, but thereby placing the crop at greater risk of cold damage (Arakawa, 1974; Tani, 1978), and at



times propelling grain prices to unprecedented levels. The famine was enhanced by the large-scale internal migration after the Shimabara Rebellion (島原の乱, Shimabara no ran, 1637–38), but also by extreme impoverishment of low-ranking members of the samurai class. Worse, a rinderpest epizootic broke out in Kyushu in 1638 and could not be contained so that mass cattle mortality occurred across Western Japan in 1640, with the consequent scarcity of working animals further reducing yields
(Atwell, 1986).

Extreme weather and poor harvests thus certainly exacerbated the disastrous situations in China and Japan. As Atwell (1986) and Zheng et al. (2014) have argued, however, a deeper understanding of the terrible "famines" of the late 1630s and early 1640s and the role of weather therein must appreciate the larger economic and social problems that had been festering in the two countries for some time, including excessive public expenditure, rapid urban growth, and intense economic competition.
These factors proved sufficiently socially and politically disruptive once that agricultural productivity could not be maintained.

**8 Final considerations**

The eruption cluster between 1636 and 1646 occurred at a time when climate, from a human perspective, had already started to worsen, and when the Maunder Minimum phase of low solar activity was about to set in (Usoskin et al., 2014; Owens et al., 2017). The eruptions also occurred in a period during which some assumed decadal effects of the 1600 CE Huaynaputina
eruption (White et al., 2021) might still have been in place. Indeed, the climatic impact of volcanic eruptions is increasingly thought to trigger dynamical responses in several internal modes of the global climate system (Robock, 2000; Timmreck, 2012), in which responses can persist on multi-decadal timescales (Zanchettin et al., 2011; Helama et al., 2021). Whether the 1600 CE Huaynaputina eruption – known for inducing massive cooling in the years following (Stoffel et al., 2015) might have triggered a cascade of North Atlantic sea ice–ocean feedbacks, and ultimately led to a persistent regime shift in ocean
circulation, an increase in Nordic Sea sea-ice extent on decadal timescales, a spin-up of the subpolar gyre and a persistent, basin-wide cooling (Lehner et al., 2013; Schleussner and Feulner, 2013) merits further investigation (White et al., 2021). If correct, cold conditions would already have prevailed over Europe at the time of the eruptions of Hekla in 1636, Koma-ga-take on 31 July 1640, Mount Parker on 4 January 1641 and Shiveluch in 1646. Contemporary historical sources and tree-ring-based reconstructions confirm the occurrence of various years of unusual cold in the early 17[th] century. Sulfur injections as
recorded in Greenland ice cores in 1636 and 1646 were likely too small to cause significant climatic perturbations alone. By contrast, volcanic gas emissions from Mount Parker in 1641 – and likely also that of Koma-ga-take in 1640 – certainly had an impact on the modes of internal climate variability as well as on (summer) temperatures (Stoffel et al., 2015; Guillet et al., 2017). However, in a context of generalized climate worsening and widespread societal conflicts at a time of "general crisis" (Atwell, 1986, 2001; Parker, 1979; Parker and Smith, 1997), and based on written and natural proxies, the direct attribution of
volcanic impacts on northern hemisphere climate – and thereafter on societal developments – is not possible without the generic assumption that correlation also means causality.



Yet it is evident that meteorological extremes – including those volcanically induced – impacted societies, particularly when survival leaned on a fragile food surplus, itself heavily predicated on the cultivation of cereals. A late European winter frost or flood could have devastating effects on summer harvests and potentially propel elevated bread prices, as in the case of the French Revolution. Thus, with a growing abundance of well-dated proxy reconstructions of climate and volcanic forcing, recent studies have increasingly examined impacts of volcanism and extreme weather on historical events and societal developments (e.g., Zhang et al., 2011; Büntgen et al., 2011, 2016, 2020; Manning et al., 2017; Campbell and Ludlow, 2020; McConnell et al., 2020; Gao et al., 2021b). Other studies have in various ways highlighted the importance of conducting such investigations with an understanding of the relevant historical and societal contexts to avoid simplistic, monocausal or environmentally deterministic linkages between extreme weather or climate anomalies and historical events (Haldon et al., 2018; van Bavel et al., 2019; Chen et al., 2020; D'Arrigo et al., 2020; Degroot et al., 2021; Guillet et al., 2021; Huhtamaa et al., 2021). Our own contribution here has thus sought to provide a summary of both the climatic and societal contexts to guide further studies.

At the same time a reactionary tendency exists to seek to minimize the role of climate (and more broadly the environment) in human affairs, whether by indiscriminately applying the label of environmental determinism to studies that acknowledge the complexity of human-environmental relationships and do not posit simplistic monocausal influences of climate on society (e.g., see Moreland's (2018) characterization of Manning et al. (2017), versus the considered assessment by van Bavel et al., 2019), or by adopting a stance (either explicitly or implicitly) in which climatic and human factors are considered in opposition as explanatory factors (e.g., Gillespie, 2020). Here we argue that the way forward is to instead recognize that all human history has taken place within socioecological systems created by continually evolving interactions with a changing environment (Ludlow et al., 2022). Embedded within this perspective is an understanding that forces such as climate and extreme weather have the effects they have because of the systems humans have created. These systems must always pay at least some regard to the prevailing environment (lest everyone starve), but major impacts can still occur when extremes are poorly adapted for, being ill-recognized or beyond experience, or deemed too costly or infrequent to merit fuller adaptation (decisions that are contingent upon prevailing societal structures, e.g., extremes may be poorly adapted to in stratified societies that insulate elites from food crises and allow other (political, religious, etc.) imperatives to take precedence.

In this manner, the climatic shocks wrought by episodes of explosive volcanism such the cluster of eruptions between 1636 and 1646 can, in the study of their impacts and the responses they provoke or fail to provoke, act as "revelatory crises" (Solway, 1994) that reveal important aspects of the underlying cultural, geopolitical and socio-economic contexts in which they occurred, and which might otherwise have remained latent (and hidden from scholars) in the absence of this sudden stress.

**Acknowledgements**

Markus Stoffel, Christophe Corona and Sébastien Guillet received funding from the Swiss National Science Foundation through the SNSF Sinergia CALDERA project (grant agreement no. CRSII5_183571). Michael Sigl received funding from the European Research Council under the European Union's Horizon 2020 research and innovation programme (grant



agreement no. 820047). Francis Ludlow received funding from the EU Horizon 2020 Marie Sklodowska-Curie CLIMCONFLICT project (grant no. 70918) and the Irish Research Council's Starting Laureate Award CLICAB project (IRCLA/2017/303). Michael Sigl received funding from the Geoscience Department and the NFR TOPPFORK project "VIKINGS" (grant no. 275191) at the University of Oslo. Chaochao Gao received funding from the Natural Science Foundation of China (grant no. 41875092). Arlene Crampsie received funding from Irish Research Council project

COALESCE/2019/43. This paper benefited from discussion facilitated by the 'Volcanic Impacts on Climate and Society' (VICS) Working Group of PAGES, funded by the Swiss Academy of Sciences and the Chinese Academy of Sciences.

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





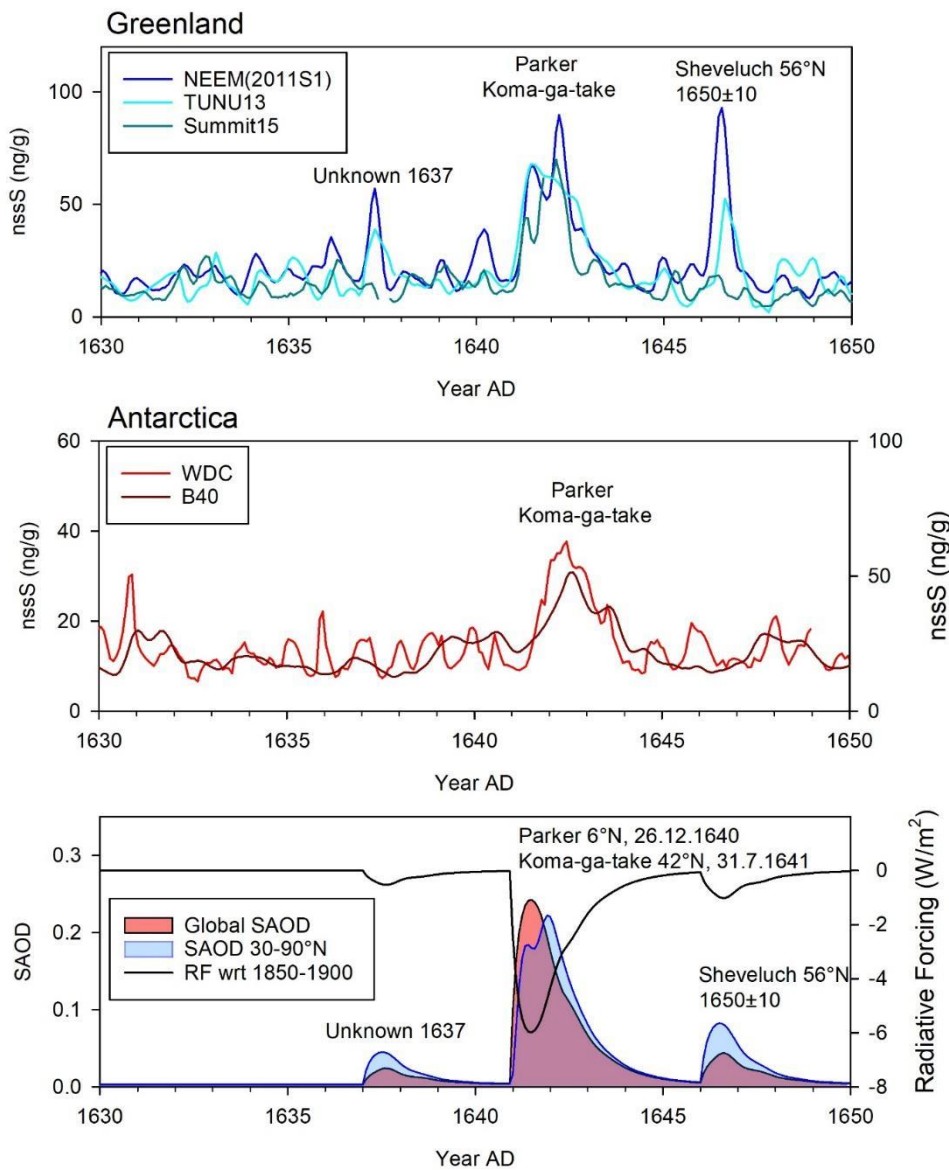

**Figure 1: Monthly resolution non-sea-salt sulfur records from ice cores in (a) Greenland and (b) Antarctica, with (c) stratospheric aerosol optical depth (SAOD) and estimated radiative forcing (RF) between 1630-1650 (data: Toohey & Sigl, 2017).**





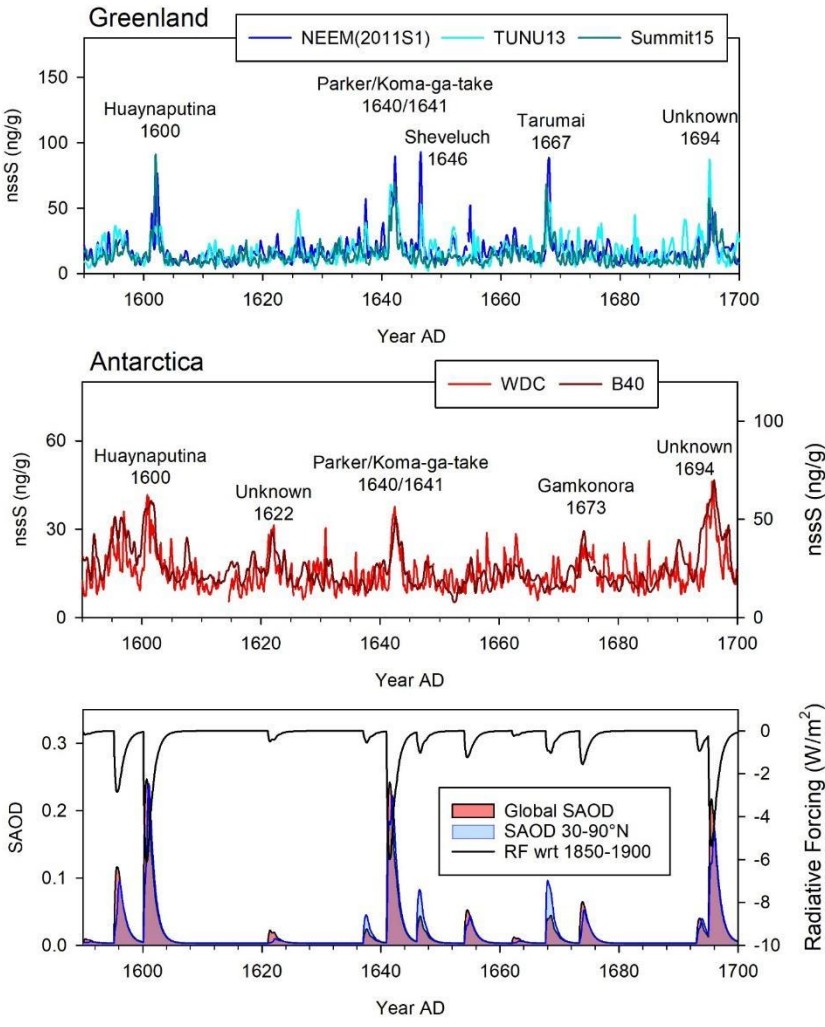

**Figure 2: Monthly non-sea-salt sulfur records from ice cores in Greenland and Antarctica, SAOD and radiative forcing during the 17th century (Toohey & Sigl, 2017).**



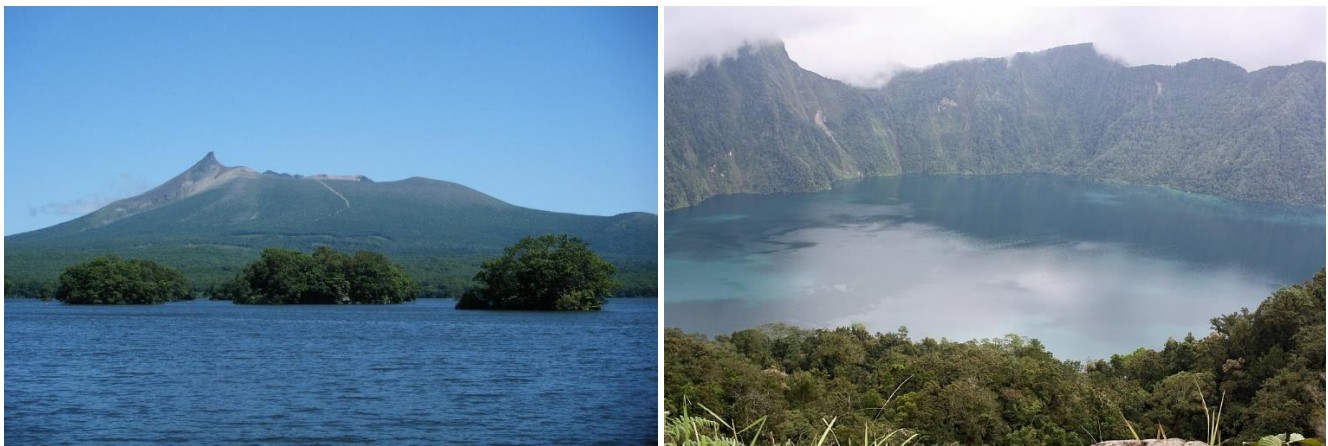

**Figure 3: (a) Koma-ga-take volcano (Hokkaido, Japan) (Source: 欅が撮影 Zelkova, CC BY-SA 3.0)  as seen from Yakumo (in the NW); (b) Mount Parker (locally known as Mélébingóy), South Cotabato, Philippines (Source: Noriah Jane Lambayan CC BY-SA 4.0).**





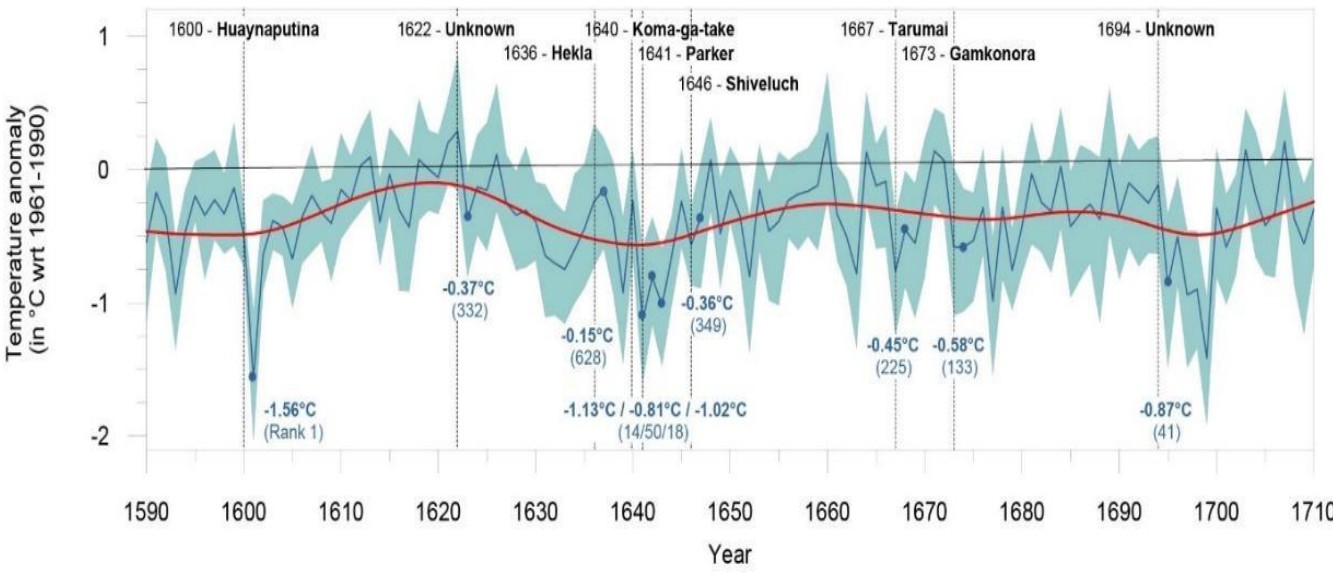


**Figure 4: Tree-ring based (NVOLC v2; Guillet et al., 2020), northern hemisphere (NH; 40-90°N) summer (JJA) temperature reconstruction of the 17th century with volcanic eruptions and the cooling induced highlighted with blue dots. The ranks indicate the amount of cooling induced by the eruption compared to all cooling events recorded over the past 1500 years. Volcanic cooling is calculated for all major 17th century eruptions: 1600 Huaynaputina, 1636 Hekla, 1640/41 Koma-ga-take / Mount Parker, 1646 Shiveluch and the 1694 unidentified eruptions.**






**Figure 5. Tree-ring reconstructions of NH extratropical land (40–90°N) summer temperature (July–August; JJA) anomalies after (a) major volcanic eruptions in 1257 (Samalas), 1600 (Huaynaputina) and 1815 (Tambora), as well as (b) between 1638 and 1643, representing the years before and after the 1640/41 Koma-ga-take and Mount Parker eruptions.**



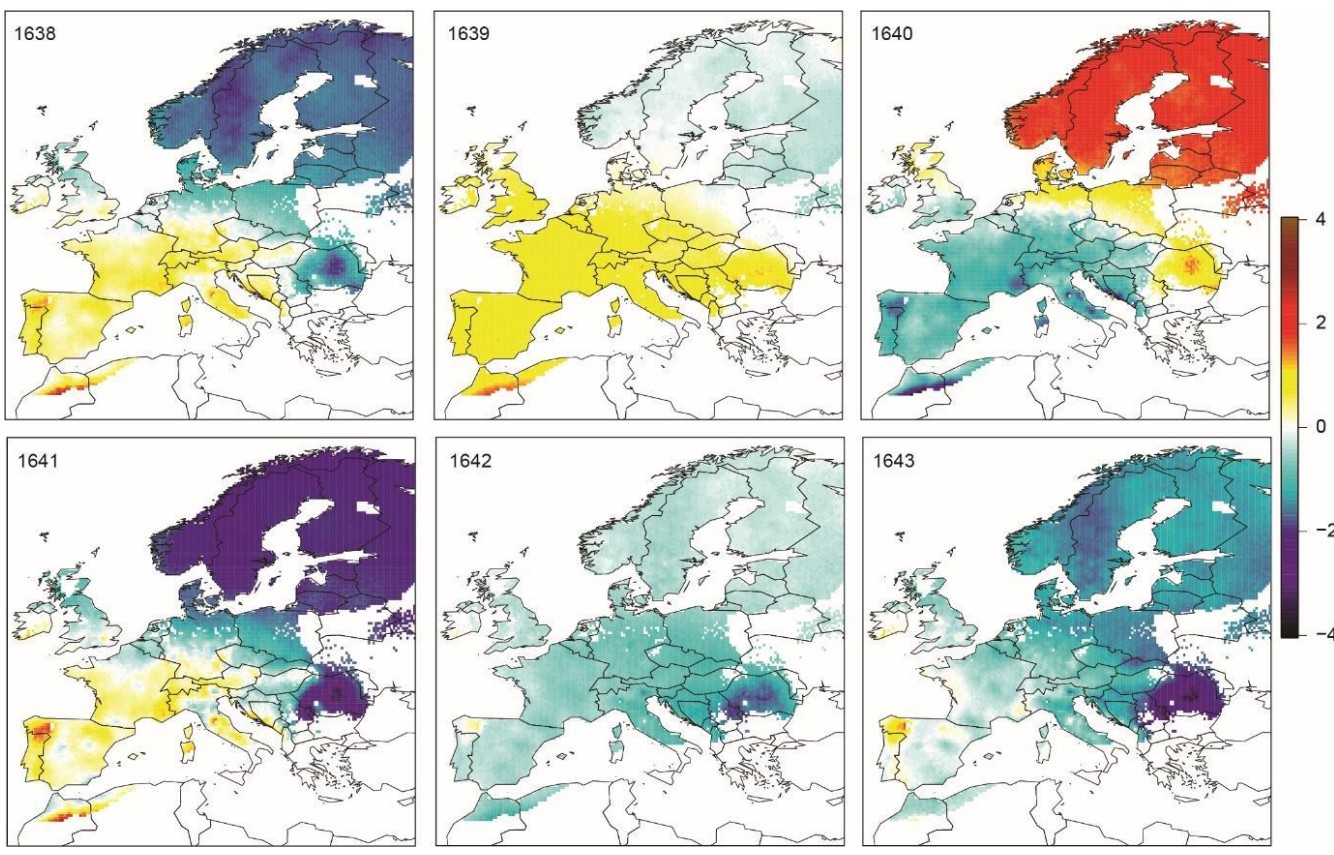

**Figure 6: Summer temperature anomalies over Europe between 1638 and 1643 as derived from a network of twelve tree-ring width, maximum latewood density and stable isotope chronologies. Temperatures are expressed as departures from summer (JJA) mean using a moving 31-yr time window.**



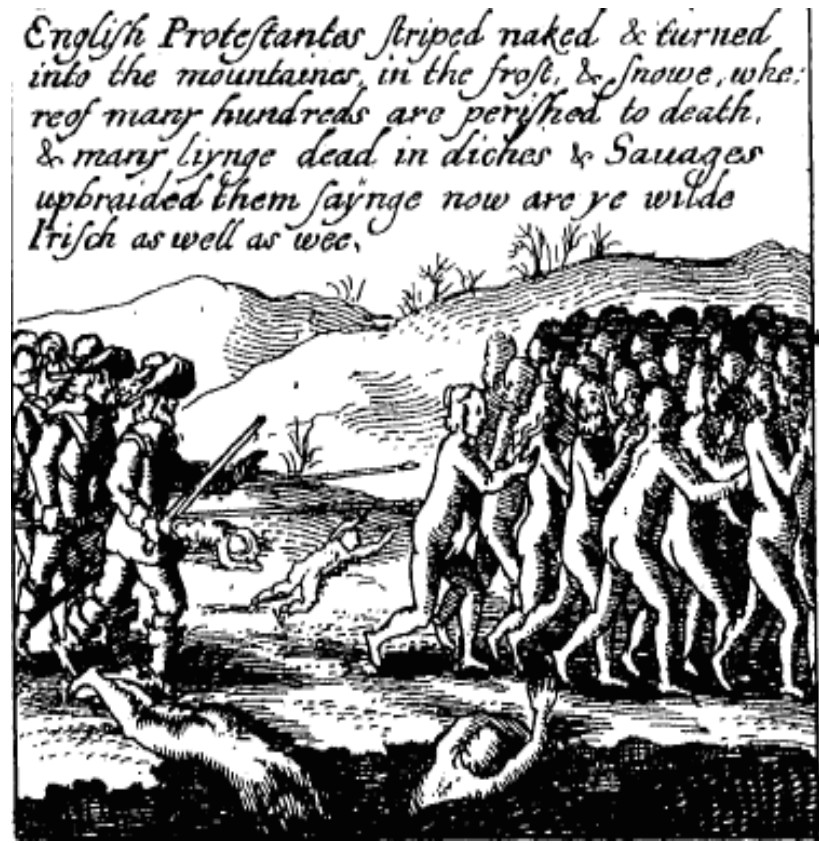

**Figure 7: Illustration (perhaps by Bohemian artist Wenceslaus Hollar) of alleged atrocities by Catholics during the winter of 1641/42; one of many accompanying the text in Presbyterian clergyman James Cranford's propagandistic The Teares of Ireland (London, 1642).**

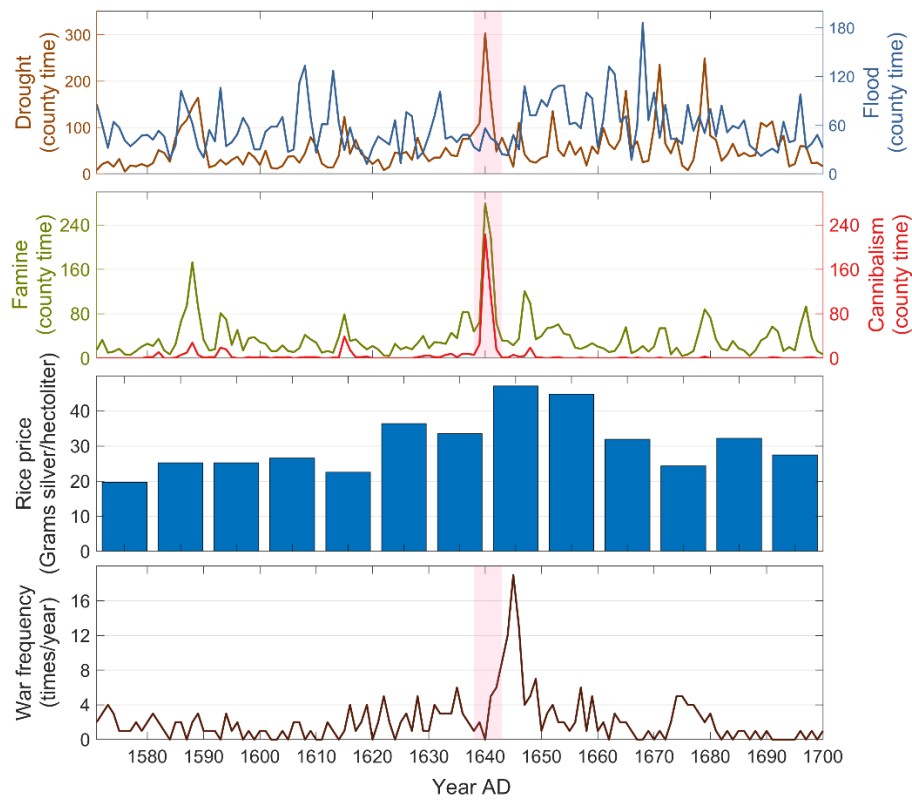

**Figure 8: Weather anomalies, food shortages and warfare, 1571 to 1700 CE: (A) drought and flood data as well as incidents of (B) famine and cannibalism as reported in the Compendium of Chinese Meteorological Records of the Last 3,000 Years (Zhang, 2004); (C) rice prices (Liu, 2015) underline the food scarcity of the early 1640s, a potential factor in favoring (D) elevated war frequencies during the period (Zhang et al., 2006).**



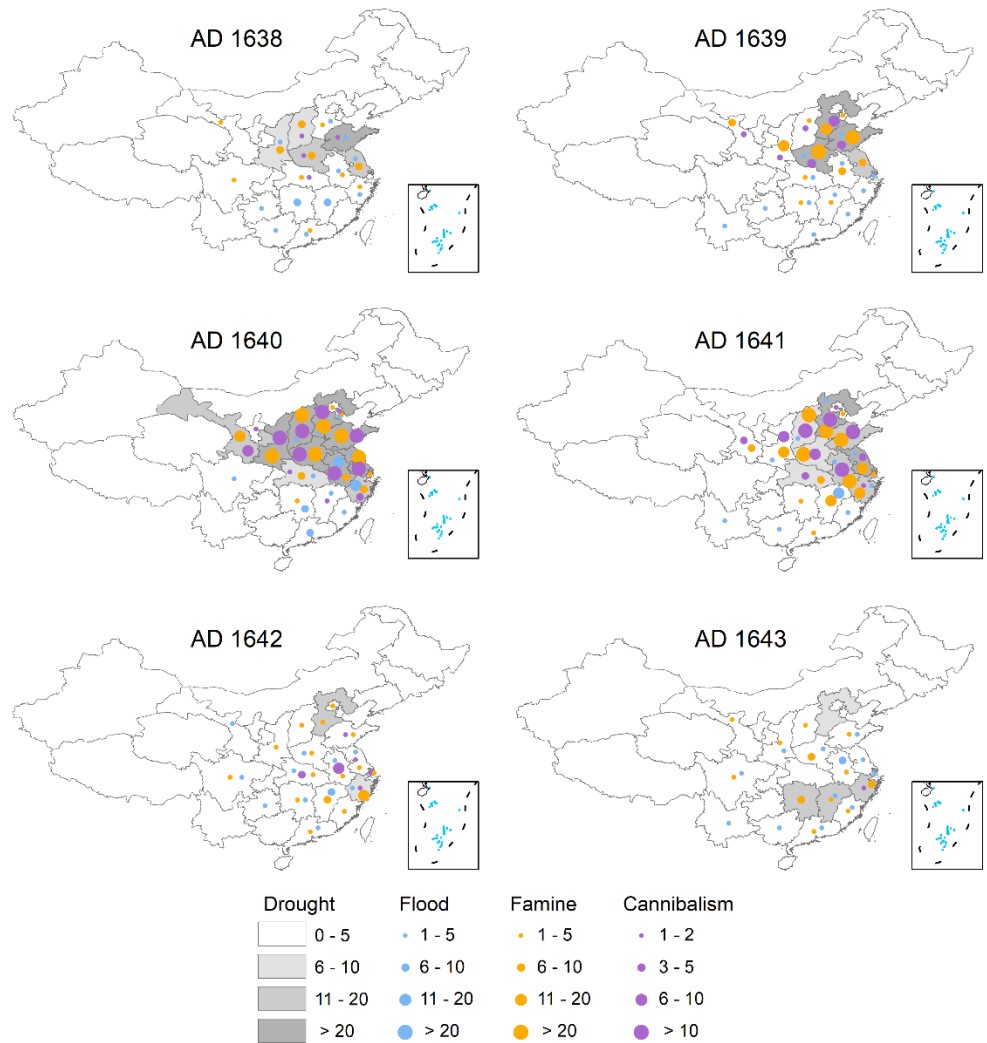

**Figure 9: Weather anomalies and social disasters, 1638 to 1643 in China. Background colors indicate recorded drought event numbers each year per province (the darker the color, the more severe the drought damage). Blues circles represent abnormal flood event frequencies each year per province (larger circles equal higher frequencies). Pink and red circles present the number of recorded famine and cannibalism events, respectively.**



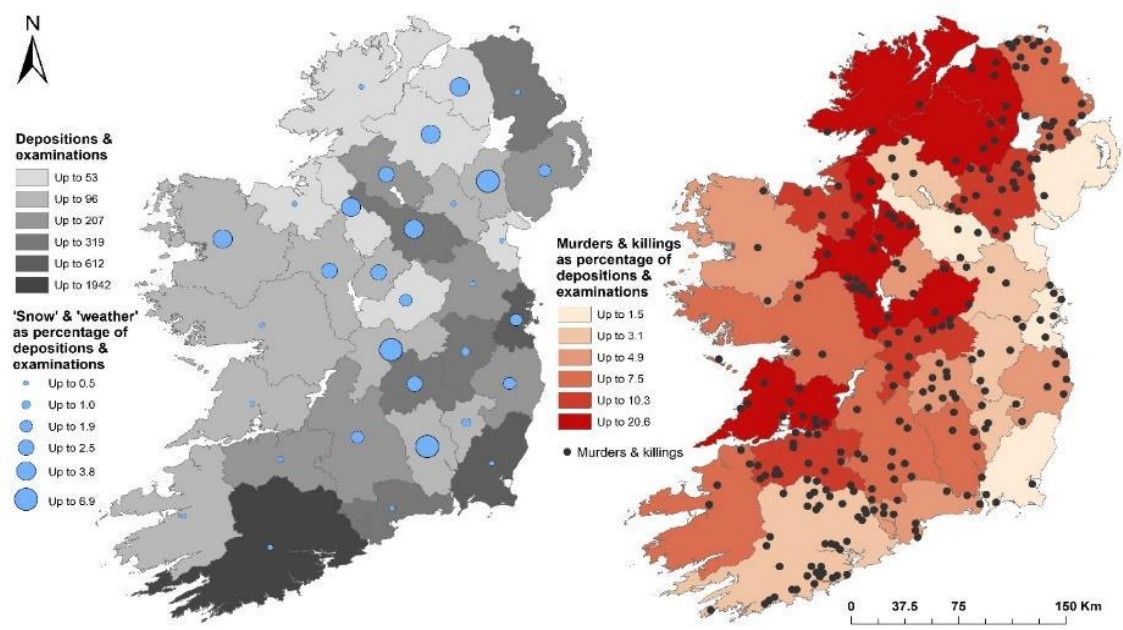

**Figure 10: Geography of reported severe weather and murders and killings during the 1641 Rebellion in Ireland. More specifically: (A) Total depositions and related documents per county (grey shading) as listed on 1641.tcd.ie (accessed 10/10/15). The percentage of these referencing 'snow' and severe 'weather' are shown in graduated blue dots; (B) Murders and killings reported in the depositions, as mapped by Smyth (2013), are shown as black dots, and expressed as a percentage of total depositions per county (red shading). Gradation into intervals in both maps is accomplished using Jenks Natural Breaks. Figure redrawn after Ludlow and Crampsie (2019).**



**Table 1:** *Metadata for volcanic eruptions and radiative aerosol forcing in the mid-17th century and for selected large historical eruptions.*

| Volcano | Pinatubo | Krakatau | Tambora | Hekla[a] | Parker | Shiveluch | Huaynaputina | Samalas |
|---|---|---|---|---|---|---|---|---|
| Region | Philippines | Indonesia | Indonesia | NH | Philippines | Kamchatka | Peru | Indonesia |
| Latitude °N | 15.1 °N | 6.2 °S | 8.3 °S | 45 °N[a] | 6.1 °N | 56.7 °N | 16.6 °S | 8.4 °S |
| Eruption Date | 02.04.1991 | 20.05.1883 | 11.04.1815 | 01.01.1637[a] | 26.12.1640 | 01.01.1646[b] | 17.02.1600 | July 1257 |
| $c_{Greenland}$ Start time (Month[c]/Year) | N/A | Sep 1883 | Jul 1815 | Jan 1637 | Feb 1641 | Feb 1646 | Jan 1601 | Mar 1258 |
| $c_{Greenland}$ Peak time (Month[c]/Year) | N/A | Apr 1884 | April 1816 | Apr 1637 | Mar 1642 | Jul 1646 | Feb 1602 | Jan 1259 |
| Duration (yrs) | N/A | 1.1 | 2.1 | 0.5 | 2.3 | 0.8 | 2.2 | 2.3 |
| $c_{Antarctica}$ Start time (Month/Year) | Jul 1991 | Jan 1884 | Jun 1815 | | Aug 1641 | | Jun 1600 | Feb 1258 |
| $c_{Antarctica}$ Peak time (Month/Year) | 1991.958 | 1885.042 | 1816.542 | | 1642.458 | | Nov 1600 | Nov 1258 |
| Duration (yrs) | 1.8 | 2.1 | 2.8 | | 2.1 | | 2.8 | 2.8 |
| $f_{Greenland}$ (kg km$^{-2}$yr) | N/A | 17.6 | 38.4 | 6.8 | 41.1 | 12.7 | 38.1 | 104.9 |
| $f_{Antarctica}$ (kg km$^{-2}$yr) | N/A | 10.4 | 45.8 | 0.0 | 14.9 | 0.0 | 18.7 | 73.4 |
| Asymmetry factor $f_{Greenland}\,/\,(f_{Greenland}+f_{Antarctica})$ | N/A | 0.63 | 0.46 | 1.00 | 0.73 | 1.00 | 0.67 | 0.59 |
| VSSI (Tg S) | N/A | 9.3 | 28.1 | 1.3 | 18.7 | 2.4 | 18.9 | 59.4 |
| Peak Time SAOD$_{30-90N}$ (Month/Year) | Mar 1992 | Mar 1884 | Jan 1816 | Jun 1637 | Nov 1641 | Jun 1646 | Dec 1600 | Jan 1258 |
| Peak SAOD$_{30-90N}$ | 0.117 | 0.139 | 0.322 | 0.045 | 0.222 | 0.083 | 0.240 | 0.580 |
| Global Peak Forcing (W m$^2$) | -2.8 | -3.3 | -8.8 | -0.5 | -6.0 | -1.0 | -6.1 | -13.9 |

*a: Unknown event tentatively assigned to Hekla and assigned eruption date of Jan 1, 1637.*

*b: Unknown event tentatively assigned to Shiveluch (1651±10 AD) and assigned eruption date of Jan 1, 1646.*

*c: Nominal month assuming linear snowfall throughout the calendar year.*



# Appendix

## Historical sources

HS1

„still trübe gegen tag nebel, NW." "dicker nebel so sich zum theil ufgezogen darauf Sonnenschein mitt einzelnen großen wolcken, E." DOI: 10.6094/tambora.org: Lenke, Walter. *Klimadaten von 1621-1650 nach Beobachtungen des Landgrafen Hermann IV. von Hessen (Uranophilus Cyriandrus)*. Offenbach a.M.: Deutscher Wetterdienst, 1960, https://www.tambora.org/index.php/grouping/event/show?event_id=106878.


HS2

"dicker nebel so sich zum theil ufgezogen darauf Sonnenschein mitt einzelnen großen wolcken, E." DOI: 10.6094/tambora.org: Lenke, Walter. *Klimadaten von 1621-1650 nach Beobachtungen des Landgrafen Hermann IV. von Hessen (Uranophilus Cyriandrus)*. Offenbach a.M.: Deutscher Wetterdienst, 1960,

https://www.tambora.org/index.php/grouping/event/show?event_id=106883

HS3

"schön warm wetter nicht recht hell, SW." DOI: 10.6094/tambora.org: Lenke, Walter. *Klimadaten von 1621-1650 nach Beobachtungen des Landgrafen Hermann IV. von Hessen (Uranophilus Cyriandrus)*. Offenbach a.M.: Deutscher Wetterdienst,

1960, https://www.tambora.org/index.php/grouping/event/show?event_id=107242.

HS4

"crassa nebula impediabat observationem" (13 August 1641). DOI: 10.6094/tambora.org: Lenke, Walter. *Klimadaten von 1621-1650 nach Beobachtungen des Landgrafen Hermann IV. von Hessen (Uranophilus Cyriandrus)*. Offenbach a.M.:

Deutscher Wetterdienst, 1960, https://www.tambora.org/index.php/grouping/event/show?event_id=107345.

HS5

"schön warm nach mitternacht nebel undt hart reif, NE." DOI: 10.6094/tambora.org: Lenke, Walter. *Klimadaten von 1621-1650 nach Beobachtungen des Landgrafen Hermann IV. von Hessen (Uranophilus Cyriandrus)*. Offenbach a.M.: Deutscher

Wetterdienst, 1960, https://www.tambora.org/index.php/grouping/event/show?event_id=107481.

HS6

"trübe heerrauchig, NW." DOI: 10.6094/tambora.org: Lenke, Walter. *Klimadaten von 1621-1650 nach Beobachtungen des Landgrafen Hermann IV. von Hessen (Uranophilus Cyriandrus)*. Offenbach a.M.: Deutscher Wetterdienst, 1960,

https://www.tambora.org/index.php/grouping/event/show?event_id=107737.



HS7

"schön wetter kalt reif, E." DOI: 10.6094/tambora.org: Lenke, Walter. *Klimadaten von 1621-1650 nach Beobachtungen des Landgrafen Hermann IV. von Hessen (Uranophilus Cyriandrus)*. Offenbach a.M.: Deutscher Wetterdienst, 1960,

https://www.tambora.org/index.php/grouping/event/show?event_id=107783.

HS8

DOI: 10.6094/tambora.org: Lenke, Walter. *Klimadaten von 1621-1650 nach Beobachtungen des Landgrafen Hermann IV. von Hessen (Uranophilus Cyriandrus)*. Offenbach a.M.: Deutscher Wetterdienst, 1960,

https://www.tambora.org/index.php/grouping/event/show?event_id=107977.

HS9

Ejusmodi notabile exemplum & mihi animadvertere contigit, Anno 25 Aprilis: Luna enim, tempore totalis obscurationis penitus evanescebat, ita ut Spectatorum haud pauci, nec locum Lunae in coelo invenire, vel indigitare potuerint; & quamvis

Telescopio instructi essemus, nihilominus visum Luna illudebat, cum tamen stellae quarti & quinti honoris, satis essent aspectabiles.

The original text can be found here: https://doi.org/10.3931/e-rara-238 (page Chapter 6, page 117)

HS10

„Im Martio ist eine solche kelte eingebrochen, dass die Wasser gefroren und lag immer schnee bis ausgangs Aprilis." Chronika oder Beschreibung deren dingen, die sich anfangs Zürcherischen Regiments bis auf dise Zeit begeben habend, geschrieben von Johann Caspar Steiner zu Zürich, Burgerbibliothek Bern, Mss.h.h.V.72-73. Quoted after Euro-Climhist.

HS11

Brunschwiler, Placidus: Diarium Fischingense 1616-1654. In: Einsiedeln, Stiftsarchiv Einsiedeln, mf 25; Chronika oder Beschreibung deren dingen, die sich anfangs Zürcherischen Regiments bis auf dise Zeit begeben habend, geschrieben von Johann Caspar Steiner zu Zürich, Burgerbibliothek Bern, Mss.h.h.V.72-73. Quoted after Euro-Climhist.

HS12

Graf, Daniel: Fragment einer Chronik. Kopie von J.J. Goldschmid In: Winterthur, Stadtbibliothek Winterthur, ms fol. 564. Quoted after Euro-Climhist.

HS13

"der Sommer war kalt, wessen etliche mahl im Julio es eys gefroren, [...]."



DOI: 10.6094/tambora.org: Dienst, Georg Andres. *Chronik von Bad Windsheim*, no year, https://www.tambora.org/index.php/grouping/event/show?event_id=103348

HS14

"1641 bey dem 21 Jul. hat jemand in seinem Calender angemerket, daß eine bis dahin um solche Zeit noch nie erlebte Kälte

eingefallen, indem in Wargen, Trutenau und andern Orten Eis gefroren." DOI: 10.6094/tambora.org: Bock, Friedrich Samuel. *Versuch einer wirthschaftlichen Naturgeschichte von dem Königreich Ost= und Westpreussen. Erster Band, welcher allgemeine geographische anthropologische, meteorologische und historische Abhandlungen enthält.* Dessau, 1782, https://www.tambora.org/index.php/grouping/event/show?event_id=106929.

HS15

"Aber keines ist dem Iahre 1641. zu vergleichen/ welches zwar auch deshalb merckwürdig gewesen/ daß die Ostern erst den 25. April eingefallen/ dergleichen in dem gantzen Seculo nicht geschehen/ aber auch von unnatürlicher Witterung keines seines gleichen gehabt; Dann ob es schon in den ersten Sommer= Monden so wohl an Feld= und Gärten= als auch Baum= Früchten wohl angelassen/ der Wein auch vor Johannis abgeblühet/ so ist doch im Monat Julio mit dem Anfange der Hunds= Tage eine

so scharfe Kälte eingefallen/ daß man hin und wieder Eiß gefunden; Es ist auch die gantze Ernde durch ein regenichtes kaltes/ wiewohl dennoch fast mit täglichen Donnern angefülltes Wetter gewesen/ welcher Gestalt alle Früchte zurücke geblieben/ und die Gerste erst nach Michaelis können gesammelt werden/ die Marellen/ oder sonsten so genannten Apricosen auch/ so sonsten eine ziemlich zeitige Sommer= Frucht sein/ haben nocht acht Tage nach Michaelis auf den Bäumen gesessen/ die spähte Baum= Früchte aber sein gar zu keiner Reife gekommen/ vielweniger der Weinwachs/ und da man wegen etlicher warmen

Tage in dem October noch Hoffnung geschöpfet/ daß er zu etwas Reife kommen würde/ so ist doch abermahl ein so starcker Frost eingefallen/ daß nicht allein die Trauben/ sondern alles junge Holtz mit einem mahle verdorben." DOI: 10.6094/tambora.org: Jobst, Wolfgang. *Johann Christoph Beckmann. Kurze Beschreibung Der Löblichen Stat Franckfurt an der Oder,* 1706 [...], https://www.tambora.org/index.php/grouping/event/show?event_id=106023.

HS16

"der Herbst fieng bald mit kalten Wetter an, den 12. und 13. Septembr. erfror der Wein Zimblich, also das Er gar sauer wurde [...]." DOI: 10.6094/tambora.org: Dienst, Georg Andres. *Chronik von Bad Windsheim*, no year, https://www.tambora.org/index.php/grouping/event/show?event_id=103340.

HS17

"1641. .. dieser Wein war ein Merkliches geringer und säurer als der serndige, weil der Weinstock in den Hundstagen auf 3 Wochen lang am Gewächs stillgestanden, hernach vor der Weinlese die Trauben fast in Berg und Tal an den Stöcken hart erfroren." DOI: 10.6094/tambora.org: Dürr, F. *Chronik der Stadt Heilbronn. Volume I: 741 – 1895.* Heilbronn:





Veröffentlichungen des Archivs der Stadt Heilbronn 27, 1986,
https://www.tambora.org/index.php/grouping/event/show?event_id=103316.

H18

"Anno 1642 im Mayen war es kalt, die Kälte that an Korn und Wein grossen Schaden." DOI: 10.6094/tambora.org: *Ulmische Chronik,* https://www.tambora.org/index.php/grouping/event/show?event_id=110298.

H19

DOI: 10.6094/tambora.org: Dienst, Georg Andres. *Chronik von Bad Windsheim*, no year, https://www.tambora.org/index.php/grouping/event/show?event_id=110286 (Bad Windsheim); Zembroth, Gallus, and Franz Joseph Mone. *Allensbacher Chronik von Gallus Zembroth*, 1863, https://www.tambora.org/index.php/grouping/event/show?event_id=110342 (Allensbach).

HS20

DOI: 10.6094/tambora.org: Gaisser, Georg Michael. *Abt Georg II. Diarien [Aufzeichnungen in Schreibkalendern Sankt Georgen Villingen]*, 1655, https://www.tambora.org/index.php/grouping/event/show?event_id=110521.

HS21

"1642 ... Den Sommer über herrschte Ruhe im Diezischen, so daß die Leute die in diesem Jahre besonders gut gediehenen Feldfrüchte ernten konnten." DOI: 10.6094/tambora.org: Heck, Robert. *Diezer Chronik oder die wichtigsten Ereignisse aus der Vergangenheit der Stadt Diez (Lahn) und ihre Dynasten (1606-1866)*, 1923, https://www.tambora.org/index.php/grouping/event/show?event_id=109736.

HS22 Maius [May 1642]. På thenna tijdhen grasserade vthi Carlstadh en häfftigh pestilentialisk febris eller en Skarck [!] brennesiuke, huilcken warade så när ett helt åhr, och vthi samma siukdom dödde månge bådhe rijke och fattige, små och stoora, the läge icke myckit länge siuke, och tå the dödde blelf therass krop all öflver medh spotter och fläckiar, och fåå voro the som icke voro siuke aff samme sott, huilcka lågo myckit länge til sängz , för ähn the förmåtte gå vppe. vthi thenne månan dödde thesse aff the förnämsta. (Petrus Magni Gyllenius, Diarium Gyllenianum eller Petrus Magni Gyllenii dagbok, edited by R. Hausen, 1822).
https://litteraturbanken.se/f%C3%B6rfattare/GylleniusP/titlar/DiariumGyllenianum/sida/68/faksimil

HS23

The course of the rebellion is detailed in thousands of interview (depositions) of the survivors taken afterwards. These depositions are housed at Trinity College and were digitized at www.1641.tcd.ie. They survive in "31 volumes, 19,010 pages,




c.3.5M words, with 8,000 witness testimonies of events relating to the 1641 rebellion. They document losses of goods and chattels, military activity, and the alleged crimes of the Irish rebels, including assault, imprisonment, the stripping of clothes,
1240 and murder, thereby providing a unique source of information for the causes and events surrounding the 1641 rebellion and for the social, economic, cultural, religious, and political history" (thank you to Jane Ohlmeyer for this summary). For the first quotation, see Ms 836, f. 89r, and the second, see MS 831, f. 77v.

HS24
1245 Spelling has been modernized. MS 836, f. 92v.

HS25
Spelling has been modernized. Deposition of Thomas Richardson, Down, MS 837, f. 013r.