# Peer review of "Climatic, weather and socio-economic conditions corresponding with the mid-17th century eruption cluster"

_Climate of the Past, 2021_

## Author Response (AR1)

Dear editor, dear Anonymous Referee #1, dear Joe Manning,

Thank you very much for the careful reading of the paper and the insightful feedback and suggestions. We would like to respond to your review below by adding our responses in italics, the replies are always proceeded by ">>>" to facilitate reading.

Thank you very much once again for the careful reading and suggestions,

Markus Stoffel, in the name of all co-authors

\*\*\*\*\*\*\*\*\*\*\*\*\*\*\*\*\*\*\*\*\*\*\*\*\*\*\*\*\*\*\*\*\*\*\*\*\*\*\*\*\*\*\*\*\*\*\*\*\*\*\*\*\*\*\*\*\*\*\*\*\*\*\*\*\*\*\*\*\*\*\*\*\*\*\*\*\*\*

RC1: Major comments:

The article by Stoffel et al. skilfully merges volcanology, palaeoclimatology and history in a convincing way. I can fully endorse its publication after some, relative minor, revisions. This multi-disciplinary team has truly put together a state-the-art-art work by assessing the (potential) impacts, and their difference in space and time, of a number of mid-17th century volcanic eruptions, some of which have attracted relatively little interest in earlier scholarship. A major strength is the inclusion of historical data from both Europe, China, and Japan. Obviously, only parts of Europe can be adequately covered. This spatial limitation could be more explicitly stated in the article. Furthermore, it would be an advantage if the treatment of China – and especially of Japan – could be more extensive. It would place Europe and Eastern Asia on a more equal footing in the article.

>>> *Thank you very much for this input. We agree completely with you that the sources that we use can at best provide a snapshot of what is available. While doing the review of existing sources – certainly incomplete –, we also tried to cover the various regions in the best possible way without making the paper longer than it already is. We suggest to address this point as follows in the revised text: "Owing to the extensive number of historical sources existing in the 17th century, this only some sources and parts of Europe and Eastern Asia can be adequately covered in the following sections."*

In the revision, the authors could check the figures and see to it that they all have the same style and fonts et cetera. For example, it appears to me as the colour scale of Figure 5 and Figure 6 differs despite that it would be better to have the same scale. Moreover, the quality of some of the figures – at least now in the pre-print – appears to have a low quality/resolution. The manuscript also needs proof-reading – this includes the reference list too – and there are inconsistencies in the list of affiliations as well (for example, the affiliation to Trinity Collage Dublin is given in two different ways).

>>> *We have been working on the style and fonts of the illustrations and also checked the quality of the illustrations. Regarding the color schemes between Figure 5 and 6, we prefer to leave them as they are as different temperature ranges are covered and therefore the scales are also different.*

>>> *We now resolved the problem of inconsistencies in the affiliation to Trinity College Dublin. Thank you for pointing this out to us.*

>>> *The reference list has been checked and inconsistencies removed. And the paper was proof-read by a native speaker.*

Minor comments:

Line 25: I am not sure if actually the term "deteriorating" is adequate here. The climate was at least as cold around 1600.

>>> *We removed the term "deteriorating"*

Line 35: What is meant with "severe" here as it cannot be a synonym to "cold" stated separately?

>>> *Agree, We now only use "cold" and have removed "severe and"*

Line 39: Is it possible to be more precise in years?

>>> *Yes, we now provide years as follows: ""*

Line 49. Also cite here: Solomina, O. N., Bradley, R. S., Jomelli, V., Geirsdottir, A., Kaufman, D. S., Koch, J., McKay, N. P., Masiokas, M., Miller, G., Nesje, A., Nicolussi, K., Owen, L. A., Putnam, A. E., Wanner, H., Wiles, G., and Yang, B.: Glacier fluctuations during the past 2000 years, Quaternary Sci. Rev., 149, 61–90, https://doi.org/10.1016/j.quascirev.2016.04.008, 2016

>>> *We added this reference.*

Lines 52–52: This cooling presumably refers to the 17th century w.r.t. 1961–1990? Not the entire multi-centennial Little Ice Age was so cold. You could alternatively cite Christiansen and Ljungqvist (2017) that the 17th century likely was 0.5–1°C colder than this reference interval. Reference: Christiansen, B., and Ljungqvist, F.C., "Challenges and perspectives for large-scale temperature reconstructions of the past two millennia"., Reviews of Geophysics, 55 (2017): 40–96. https://doi.org/10.1002/2016RG000521

>>> *Thank you for this valuable suggestion We have changed the number and added the reference.*

Line 61: I presume that you with the "Little Ice Age" here mean the late 17th century only? It should be rephrased.

>>> *That's true, we rephrased the sentence as follows: ""*

Line 66: "Thirty Years War" should be "Thirty Years' War".

>>> *Done*

Line 68: A better reference than Parker (2013) here is Parker (2006): Parker, G., 2006. The Thirty Years' War. Routledge, London.

>>> *Thank you, we adjusted the reference accordingly.*

Lines 71–72: Is "Portugal rebels" right here?

>>> *We have changed the text as follows as the original text may not have been clear: "Moreover, the period was characterized by civil wars including the Scottish Revolution (1637–1644), the Croquant (1637) and the Nu-pieds (1639) revolts in France, the Catalan Revolt (1640–1659), the Portuguese Revolution (1640, continuing to 1668 as the War of Restoration), the Irish Rebellion (1641–1642) and English Civil War (1642–1651) (Parker 2013)."*

Line 118–120: This very short paragraph could be merged with another paragraph.

>>> *Agree, done*

Line 249: "Jamtland" should be "Jämtland".

*>>> Changed*

Line 271: Unclear what eruption "rank 4" refers to here.

*>>> rank 4 signifies simply that 1633 was the fourth-coldest year in the reconstruction. It occurred in the absence of volcanic activity.*

Line 315: Consider to rephrase this slightly. Catholic France fought on the "Protestant" side.

*>>> Done*

Line 348: As grain price data are available for Paris, maybe you could quantify the price increase?

*>>> We have added more information as follows: "In neighboring France, the grape harvest between 1640 and 1643 began a full month later than usual and wheat prices surged from 12 to 20 pounds (per setier, that is about 7 liters) between 1641 and 1642 (Baulant, 1968), indicating poor cereal harvests, especially also in eastern France.".*

Line 362: "arrival" is better here than "rise". And "bulk" better than "heavy".

*>>> Thank you for these suggestions, we have changed the text accordingly*

Line 395: Maybe a citation to some work by Astrid E. J. Ogilvie is better here than to Parker (2013)

*>>> We have added Ogilvie's 1984 chapter in the Reidel (Dordrecht) book edited by N.-A. Morner and W. Karlen (Climate changes on a yearly to millennial basis)*

Line 508: Other estimates of the number of deaths in the Thirty Years' War are even higher. See, e.g.: Wilson, Peter H. (2009). Europe's Tragedy: A History of the Thirty Years War. Allen Lane.

*>>> Thank you for this feedback. We have adjusted the numbers accordingly.*
* * *
RC2 (Joe Manning)

*>>> Thank you very much for the feedback on the paper and the kind words. We have added the reference of Drixler (2013) referring to infanticide in Japan and changed African to East African Monsoon.*

---

## Author Response (AR2)

Dear editor, dear Jürg,

Thank you for your message and the request for minor revision which we received on March 10, 2022. Below you will find a point-by-point reply:

EDITOR: lines 365 and 1170 contain comments of Francis, they should be clarified.

>>> *Thank you for the comment and our apologies for the omission. The comment in line 365 has now been removed and the new version is clean. The sentence reads as follows: "Paris has excellent climate observations between 1630 and 1640 in municipal acts" as it indeed refers to the period spanning from 1630 to 1640. In line 1170, hyphens have been adjusted and are now the same throughout the document.*

EDITOR: Also line 1130 in fig caption 6 has a comment of Markus that should deleted.

>>> *The comment has now been deleted.*

EDITOR: Figs 5 and 9 need references in the caption.

>>> *Agree, we have added "NVOLC v2; Guillet et al., 2020" to Fig. 5 and "Gao et al., 2017" to Fig. 9.*

[Figure]

**Figure R1: Temperature reconstruction (1638 – 1643) based on the Luterbacher et al. (2004) dataset**

EDITOR: Within the section of Figure 6, please compare with the summer reconstructions published in Luterbacher et al. 2004 and add a few sentences if they are in agreement or not and add their maps in Fig 6 so that the reader can compare. Otherwise the reader gets the impression, that your reconstructions are the first ones for those years

*>>> We have compared our summer reconstruction (0.25 x 0.25° grid) which is based on 12 MXD, TRW and isotope chronologies with the dataset published with by Luterbacher et al. (2004). The temperature reconstructions based on the Luterbacher dataset are presented below. This dataset, published in 2004, contains primarily the climate indices of Christian Pfister (Switzerland), Rüdiger Glaser (Germany) and van Engelen (Low Countries) as well as TRW data from Lofoten Island and Yamal, as well as discontinuous data from Czech Republic and Hungary, and is resolved at 0.5 x 0.5°.*

*We observe major differences between our reconstruction and the Luterbacher data (see Fig. R1). We explain these differences by the more limited number of summer-proxy data used and somewhat more limited representativeness of the Lofoten and Yamal reconstructions for summer temperatures in Central and Western Europe. In addition, the climate indices provided by historians (Glaser, Pfister, van Engelen) – while certainly very valuable – are semi-quantitative and based on ordinal-scale climate indices.*

*By contrast, the dataset that we use in the paper contains a substantially larger number of tree-based records across Europe; many of these datasets were not available at the time of publication of the Luterbacher record in 2004.*

*We also validated our tree-based reconstruction against a spatially-explicit, dense network of independent summer-temperature sensitive grape harvest records (Fig. R2) for Central and Western Europe (Daux et al., 2012; Corona et al., in review). Whereas the grape harvest reconstruction agrees very well with the tree-based reconstruction presented in this study, it diverges as well from the Luterbacher et al. (2004) dataset. We conclude that for the early- to mid-17th century, differences in the number and spatial representation of proxies best explains the different results obtained.*

[Figure]

**Figure R2: Temperature reconstruction (1638–1643) based on grape harvest dates (Daux et al., 2012, Corona et al., under review)**

*We therefore decided not to include the Luterbacher reconstruction in the paper. Instead, we added a sentence in the manuscript in lines 260-1 to explain the choice of the dataset used in this paper: "The dataset that we use in this study contains a larger number of summer-temperature sensitive proxies – especially over Western Europe and Scandinavia – and has a finer resolution than the Luterbacher et al. (2004) dataset.*

We look forward to hearing from you and to see this paper published soon.

Kind regards

Markus Stoffel, in the name of all co-authors

**References**

Corona et al., Comm Earth & Environ, under review.
Daux et al., Clim. Past, 8, 1403–1418, 2012.
Luterbacher et al., Science 303, 1499–1503, 2004

---

## Author Response (AR3)

Dear editor, dear Jürg,

Thank you for your message and the request for minor revision which we received on March 31, 2022. We look forward to hearing from you and to see this paper published soon.

We have added two paragraphs and an illustration combing our with the Luterbacher et al. (2004) datasets. All changes are highlighted in yellow.

Kind regards

Markus Stoffel, in the name of all co-authors